# Sensorless Fractional Order Control of PMSM Based on Synergetic and Sliding Mode Controllers

## Marcel Nicola [1,*] and Claudiu-Ionel Nicola [1,2,*]

[1] Research and Development Department, National Institute for Research, Development and Testing in Electrical Engineering—ICMET Craiova, 200746 Craiova, Romania

[2] Department of Automatic Control and Electronics, University of Craiova, 200585 Craiova, Romania

* Correspondence: marcel_nicola@yahoo.com (M.N.); claudiu@automation.ucv.ro (C.-I.N.)

**Abstract:** The field oriented control (FOC) strategy of the permanent magnet synchronous motor (PMSM) includes all the advantages deriving from the simplicity of using PI-type controllers, but inherently the control performances are limited due to the nonlinear model of the PMSM, the need for wide-range and high-dynamics speed and load torque control, but also due to the parametric uncertainties which occur especially as a result of the variation of the combined rotor-load moment of inertia, and of the load resistance. Based on the fractional calculus for the integration and differentiation operators, this article presents a number of fractional order (FO) controllers for the PMSM rotor speed control loops, and $i_d$ and $i_q$ current control loops in the FOC-type control strategy. The main contribution consists of proposing a PMSM control structure, where the controller of the outer rotor speed control loop is of FO-sliding mode control (FO-SMC) type, and the controllers for the inner control loops of $i_d$ and $i_q$ currents are of FO-synergetic type. Superior performances are obtained by using the control system proposed, even in the case of parametric variations. The performances of the proposed control system are validated both by numerical simulations and experimentally, through the real-time implementation in embedded systems.

**Keywords:** permanent magnet synchronous motor; fractional order control; synergetic control; sliding mode control

## 1. Introduction

The permanent magnet synchronous motor (PMSM) is widely used in industrial applications, the aerospace industry, electric vehicles, robotics, electric drives and computer peripherals. The popularity of using the PMSM for a very wide range of applications is due to a set of advantages such as efficiency, small size, high power and high torque density. Naturally, for the control of the PMSM, a number of algorithms and control strategies have been developed, both in the range of the classic type of control, and also as through modern and unconventional approaches. The field oriented control (FOC) and direct torque control (DTC) [1–7] can be distinguished among the control strategies of the PMSM. The DTC strategy is characterized by a simpler structure in terms of controllers which are generally ON-OFF, but inherently the performance of the control system is affected by the occurrence of oscillations. The FOC strategy contains a cascade control structure, where the outer loop controls the PMSM rotor speed, and the inner control loops control currents $i_d$ and $i_q$. In the classical approach, the FOC strategy controllers are PI type. This approach includes all the advantages provided by the simplicity of using such controllers, but inherently the control performances are limited due to the nonlinear model of the PMSM, the need for wide-range and high-dynamics speed and load torque control, but also due to the parametric uncertainties which occur especially as a result of the variation of the combined rotor-load moment of inertia, and of the load resistance.

Among the more complex control systems used to obtain superior performances we can mention the adaptive control [8–10], the predictive control [11–13], the robust control [14,15], the backstepping control [16], the sliding mode control (SMC) [17,18] and the synergetic control [19,20]. These types of control systems provide superior performance in terms of the response time, the overshoot and the parametric robustness, and the real-time implementation in embedded systems can be achieved with digital signal processors (DSP) with common performance, with a very good performance/price ratio.

Among these control systems, due to their robustness to parametric variations, the SMC-type control systems have a special role, as well as the development of low-order controllers, with obvious advantages in the real-time implementation in embedded systems. To counter SMC's main disadvantage due to the occurrence of the chattering phenomenon, a series of techniques have been developed, among which we mention the use of a proportional-integral type of sliding surface plus integrator or a second-order SMC. Further, we recall that the synergetic control can be considered as a generalization of the SMC-type control, retaining the decoupling and model order reduction properties of this type of control and the advantages provided by this approach for the synthesis of the controller.

We can mention the following intelligent control systems: fuzzy [21], neuro-fuzzy [22], artificial neural network (ANN) [23–25], particle swarm optimization (PSO) or genetic algorithms [26]. These types of control also provide superior performance, but in the real-time implementation in embedded systems it is necessary to use very fast DSPs, with relatively high costs, in addition to a number of specific software libraries of the control application development environments. Thus, the performance/price ratio does not recommend the widespread use of these types of controllers.

Regarding the elimination of the speed sensors, in order to increase the reliability of the system, Luenberger [27], model reference adaptive system (MRAS) [28], and sliding mode observer (SMO) speed observers [29,30] are used for the deterministically described systems, and Kalman type observers [31] are used for the stochastic description of the system. Evidently, according to the performance-cost criterion, the deterministic observers are the most commonly used. Furthermore, a range of observers have been developed for the detection of faults, one of the most useful observers being used for the detection of faults in current sensors on the supply phases of the PMSM.

One of the special applications of using the fractional calculus for integration and differentiation operators consists of obtaining the fractional order (FO) controllers [32–35], in order to obtain superior control performance. In this sense, the first approaches were obviously aimed at obtaining FO-PI-type controllers. Although the development of the FO-type controllers is very attractive due to the possibility for finer tuning of certain tuning parameters, which are traditionally integer and fixed parameters (for example the power of operator s in the structure of the PI or PID-type controller), the study of these controllers was greatly accelerated with the development of specialized toolboxes such as the fractional order modeling and control (FOMCON) integrated into the MATLAB/Simulink environment.

Among the usual applications of the PMSM control systems we mention: maintaining the speed according to a profile set by a speed reference generator, but also master/slave type multi-motor applications where the coupling is rigid or flexible and it is necessary to maintain the same speed or maintain the torque developed by each engine in the narrowest range possible [36]. Furthermore, the electric vehicles drive control applications raise the problem of multi-motor speed control [37]. Applications such as the automatic control of the hydropower dam spillway require that the error accumulated in each drive chain corresponding to each engine be less than the set value [38]. These applications are generally achieved using the controllers described above, but of the integer order type. In this article we will focus on the fractional order controllers which provide superior control performance, but also on the increased difficulties regarding the implementation in embedded systems.

This article compares the performances obtained using FO-PI, tilt integral derivatives (TID), FO-lead lag controller, and SMC speed controllers against the classic PI-type speed controller in an FOC-type control structure of the PMSM, under the conditions where the controller of the current loops is of PI type. It also presents the performances obtained by using the synergetic control for the control of currents $i_d$ and $i_q$, within an FOC-type control structure of the PMSM with PI speed controller.

The main contribution consists of proposing a PMSM control structure, where the controller of the outer rotor speed control loop is of FO-SMC type, and the controllers for the inner control loops of $i_d$ and $i_q$ currents are of FO-synergetic type. Superior performances are obtained by using the control system proposed, even in the case of parametric variations. The FO-SMC controller outputs the current reference $i_{qref}$, while $i_{dref} = 0$ according to the FOC control strategy. The FO-synergetic-type controllers directly provide the control inputs $u_d$ and $u_q$, and the control of the inverter is performed through the inverse Park and Clarke transformations from d-q reference frame system to abc reference frame system. The validation of the results presented is achieved by numerical simulations, but also by real-time implementation in embedded systems.

The rest of the paper is organized as follows: the basic concepts of the fractional calculus for integration and differentiation operators are presented in Section 2, the FOC-type control strategy and the transfer function of the PMSM are presented in Section 3. The fractional order speed controllers for the PMSM are presented in Section 4, Section 5 presents the fractional order synergetic current controllers for the PMSM and Section 6 presents the observers for rotor speed estimation and fault detection. Sections 7 and 8 present the numerical simulations and the experimental results, respectively. Some conclusions are presented in the last section.

## 2. Fractional Order Calculus

Let us note that the non-integer order operator for integration and differentiation as $aD_t^\alpha$, where $\alpha$ represents the fractional order, $a$ and $t$ represent the limits of the interval at which the operator is applied [27,28].

$$aD_t^\alpha = \begin{cases} \frac{d^\alpha}{dt^\alpha} & \text{Re}(\alpha) > 0 \\ 1 & \text{Re}(\alpha) = 0 \\ \int_a^t (dt)^{-\alpha} & \text{Re}(\alpha) < 0 \end{cases} \tag{1}$$

Although there are several ways to define this operator, not all have a generally accepted meaning. The most widely used definition is the Riemann–Liouville differintegral [32,33]:

$$aD_t^\alpha f(t) = \frac{1}{\Gamma(m-\alpha)} \left(\frac{d}{dt}\right)^m \int_\alpha^t \frac{f(\tau)}{(t-\tau)^{\alpha-m+1}} d\tau \tag{2}$$

where $m - 1 < \alpha < m$, $m \in N$, and $\Gamma(\cdot)$ represents Euler's gamma function. Another useful definition in practical applications is given by Grünwald–Letnikov [32,33]:

$$aD_t^\alpha f(t) = \lim_{h \to 0} \frac{1}{h^\alpha} \sum_{j=0}^{\left(\frac{t-\alpha}{h}\right)} (-1)^j \binom{\alpha}{j} f(t - jh) \tag{3}$$

where $(\cdot)$ represents the integer part.

Similarly to the case of the integer order of the operator defined in (1), the Laplace transform is applied and the transfer function for signals and with fractional derivative is defined. For example, if the orders of the fractional operator s are integer multiples in relation to the commensurate order $q$, $(q \in R^+, 0 < q < 1, \alpha_k = kq)$, the transfer function $H(\lambda)$ can be expressed as follows:

$$H(\lambda) = \frac{\sum_{k=0}^{m} b_k \lambda^k}{\sum_{k=0}^{n} a_k \lambda^k} \tag{4}$$

where $\lambda = s^q$.

However, in the case of linear and time-invariant systems, in the fractional case, the state space representation becomes:

$$D^q x(t) = Ax(t) + Bu(t)$$
$$y(t) = Cx(t) + Du(t)$$
(5)

Furthermore, the stability of the system (5) can be verified by fulfilling the following relationship:

$$|\arg(eig(A))| > q\frac{\pi}{2}$$
(6)

where $0 < q < 1$ represents the commensurate order, and $eig(A)$ is the eigenvalue of the associated matrix $A$.

To obtain a good approximation of a transfer function with fractional order in a specified frequency range $(\omega_b, \omega_h)$ and of order N, Oustaloup's recursive filter for $s^\gamma$ and $0 < \gamma < 1$ can be used as follows [32,33]:

$$G_f(s) = K \prod_{k=-N}^{N} \frac{s + \omega_k'}{s + \omega_k}$$
(7)

where $\omega_k'$, $\omega_k$ and $K$ are given by:

$$\omega_k' = \omega_b \left(\frac{\omega_h}{\omega_b}\right)^{\frac{k+N+\frac{1}{2}(1-\gamma)}{2N+1}} ; \; \omega_k = \omega_b \left(\frac{\omega_h}{\omega_b}\right)^{\frac{k+N+\frac{1}{2}(1+\gamma)}{2N+1}} ; \; k = \omega_h^\gamma$$
(8)

Furthermore, a refined Oustaloup filter is given by [32]:

$$s^\alpha \approx \left(\frac{d\omega_h}{b}\right)^\alpha \left(\frac{ds^2 + b\omega_h s}{d(1-\alpha)s^2 + b\omega_h s + d\alpha}\right) G_p$$
(9)

$$G_p = K \prod_{k=-N}^{N} \frac{s + \omega_k'}{s + \omega_k}; \; \omega_k = \left(\frac{b\omega_h}{d}\right)^{\frac{\alpha+2k}{2N+1}}; \; \omega_k' = \left(\frac{d\omega_b}{b}\right)^{\frac{\alpha-2k}{2N+1}}$$
(10)

In Equations (9) and (10), usually $b = 10$ and $d = 9$.

## 3. Mathematical Model of PMSM. Transfer Function Representation. FOC Strategy of PMSM

The mathematical model of the PMSM in the rotor reference frame (d-q frame) by applying the Park transform and according to [1,2] is obtained in the following form:

$$\begin{bmatrix} u_q \\ u_d \end{bmatrix} = \begin{bmatrix} R_q + \rho L_q & \omega_e L_d \\ -\omega_e L_q & R_d + \rho L_d \end{bmatrix} \begin{bmatrix} i_q \\ i_d \end{bmatrix} + \begin{bmatrix} \omega_e \lambda_0 \\ \rho \lambda_0 \end{bmatrix}$$
(11)

where $u_d$, $u_q$ and $i_d$, $i_q$ are the stator voltages and currents in the d-q reference frame of the PMSM, $L_q$, $L_d$ and $R_q$, $R_d$ are the stator inductances and resistances in the d-q reference frame, $\omega_e$ is the electrical angular velocity of the rotor, $\lambda_0$ is the flux linkage, and $\rho$ is the differential operator.

The flux on the d-q axes is expressed as:

$$\lambda_q = L_q i_q$$
$$\lambda_d = L_d i_d + \lambda_0$$
(12)

By indicating the electromagnetic torque developed by the PMSM as $T_e$, the following relations on the PMSM dynamics can be expressed:

$$T_e = \frac{3}{2} n_p \left(\lambda_d i_q - \lambda_q i_d\right); \; T_e = K_t i_q$$
$$T_e = T_L + B\omega + J\frac{d\omega}{dt}$$
(13)

where $K_t = \frac{3}{2}n_p\lambda_0$ represents the torque constant, $B$ represents the viscous friction coefficient, $J$ represents the rotor inertia, $n_p$ represents the number of pole pairs, and $T_L$ represents the load torque.

By assuming the following simplifications $L_d = L_q = L$, $R_d = R_q = R_s$, and $\omega_e = n_p \cdot \omega$, where $\omega$ is the angular velocity of the rotor, the following PMSM model can be obtained:

$$\begin{pmatrix} \dot{i}_d \\ \dot{i}_q \\ \dot{\omega} \end{pmatrix} = \begin{pmatrix} -\frac{R_s}{L} & n_p\omega & 0 \\ -n_p\omega & -\frac{R_s}{L} & -\frac{n_p\lambda_0}{L} \\ 0 & \frac{K_t}{J} & -\frac{B}{J} \end{pmatrix} \begin{pmatrix} i_d \\ i_q \\ \omega \end{pmatrix} + \begin{pmatrix} \frac{u_d}{L} \\ \frac{u_q}{L} \\ -\frac{T_L}{J} \end{pmatrix} \tag{14}$$

Using the Equations (11)–(14) describing the PMSM, Figure 1 shows the block diagram by reduced transfer functions of the PMSM rotor speed control system. In addition to the notations presented above, we denote the transfer functions of current and speed sensors as $H_c(s)$ and $H_\omega(s)$ in Figure 1. Usually these are 1st order transfer functions where the time constant is in the order of milliseconds, thus allowing a simplified approach to the reduction of these transfer functions to constants.

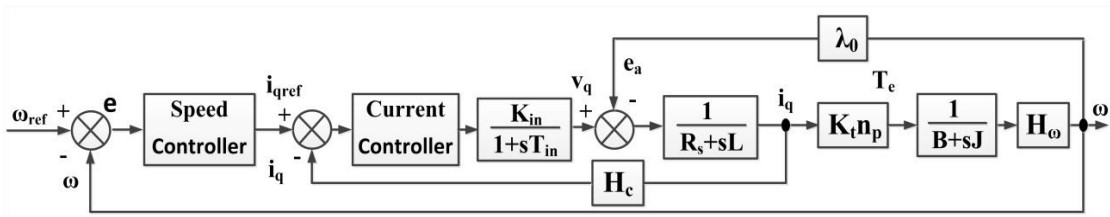

**Figure 1.** The block diagram of the speed loop of the permanent magnet synchronous motor (PMSM) drive.

The transfer function of the inverter is as follows:

$$H_{in} = \frac{K_{in}}{1 + sT_{in}} \tag{15}$$

where: $K_{in} = 0.65 \cdot (V_{dc}/V_{cm})$ and $T_{in} = 1/(2 \cdot f_c)$, $V_{dc}$ represents the dc link voltage (input of the inverter), $V_{cm}$ represents the maximum control voltage, and $f_c$ represents the switching frequency of the inverter.

In Figure 1 by shifting the point of intersection of the back-electromotive force (back-EMF) loop with the speed loop to the point of intersection with the current loop, an equivalent form of defining the fixed part (the current control loop of $i_q$) as transfer functions is shown in Figure 2 [39].

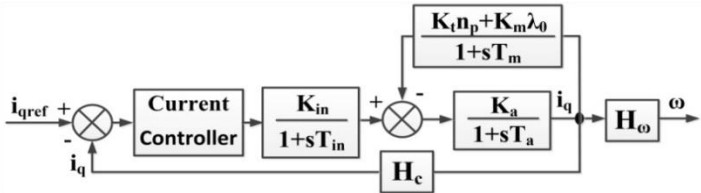

**Figure 2.** The block diagram of the current control loop of the PMSM drive.

In Figure 2, using the following notations:

$$\begin{cases} K_a = \frac{1}{R}; \ T_a = \frac{L}{R}; \ K_m = \frac{1}{B}; \ T_m = \frac{J}{B}; \ K_b = K_tK_m\lambda_0; \\ T_i = 0.1T_m; \ K_{in} = 20; \ K_i = (T_mK_{in})/(T_2K_b) \end{cases} \tag{16}$$

is the transfer function of the inner current loop which represents the fixed part of the speed control system (outer control loop) which becomes as follows:

$$H_f = \frac{K_t K_i}{(as^3 + bs^2 + cs + d)} \tag{17}$$

where

$$\begin{cases} a = T_i LJ; \ b = LJ + T_i(LB + R_s J) \\ c = LB + R_s J + T_i(R_s B + \lambda_0 K_t); \ d = R_s B + \lambda_0 K_t \end{cases} \tag{18}$$

Figure 3 shows the proposed general block diagram of the enhanced FOC control type strategy for PMSM. In this paper, the speed controller of the outer control loop is a classic PI-type controller, but, as shown in Section 4, it can be replaced with FO-PI, TID, FO-lead-lag, and FO-SMC controllers. Furthermore, compared to the classic approach, the PI-type current controllers in the inner control loop can be replaced with synergetic and FO-synergetic controllers, as presented in Section 5. The output of these controllers provides the reference $i_{qref}$ for the inner current control loop, where $i_{dref}$ = 0. Section 6 presents improvements that can be made to the classic FOC-type scheme by using an SMO-type observer to estimate the rotor position and speed, but also an FDO observer to detect the faults on the PMSM supply phases.

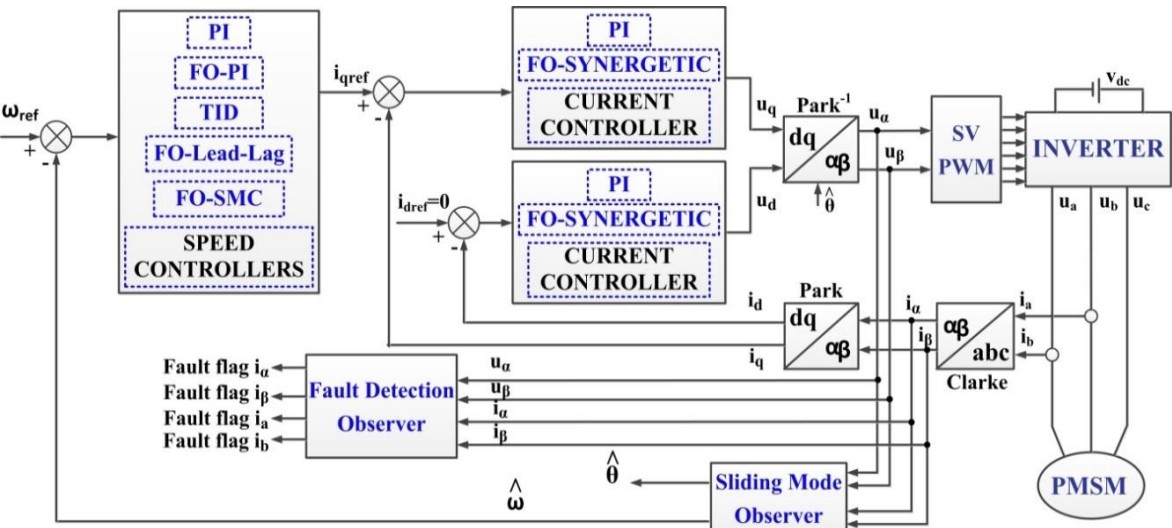

**Figure 3.** Enhanced field oriented control (FOC) control strategy for the PMSM—general block diagram.

## 4. Fractional Order Speed Controllers for PMSM

By using the fractional-type control, whose basic elements were described in Section 2, this section presents the equations of certain fractional-type controllers that will replace the PI-type speed controller in the FOC-type control strategy of PMSM.

### 4.1. FO-PI Speed Controller

Furthermore, in terms of the fractional controllers, the most commonly used are the $PI^\lambda D^\mu$ controllers, which can be expressed as follows [33]:

$$u(t) = K_p e(t) + K_i D^{-\lambda} e(t) + K_d D^\mu e(t) \tag{19}$$

where *e(t)* represents the error signal.

After applying the Laplace transform to Equation (19), by assuming zero initial conditions, the following equation is obtained:

$$G_c(s) = K_p + \frac{K_i}{s^\lambda} + K_d s^\mu \tag{20}$$

where $K_p$ represents the proportional gain, $K_i$ represents the integrator gain, $\lambda$ represents the integrator order (positive), $K_d$ represents the differentiator gain, and $\mu$ represents the differentiator order. For $\lambda = \mu = 1$, the result is the usual integer-order PID controller.

### 4.2. TID Speed Controller

Another fractional controller used in applications is the TID controller, which can be described by the next transfer function [33]:

$$G_c(s) = \frac{K_t}{s^{1/n}} + \frac{K_i}{s} + K_d s \tag{21}$$

where $K_t$ represents the tilt gain, $n$ represents the tilt integrator order, $K_i$ represents the integrator gain, and $K_d$ represents the differentiator gain.

### 4.3. Lead-Lag Speed Controller

The general form of the transfer function of an FO-lead-lag controller is given by [33]:

$$G_c(s) = K_c \left( \frac{s + \frac{1}{\lambda}}{s + \frac{1}{x\lambda}} \right)^{\alpha} = K_c x^{\alpha} \left( \frac{\lambda s + 1}{x\lambda s + 1} \right)^{\alpha}, \quad 0 < x < 1 \tag{22}$$

where $\lambda$ represents the fractional order of the FO-lead-lag controller.

It is noted that, for $\alpha > 0$, a lead effect of the FO-lead-lag controller is obtained, while for $\alpha < 0$, a lag effect of the FO-lead-lag controller is obtained.

For $k' = K_c x^{\alpha}$, the common form of the FO-lead-lag controller is obtained:

$$G_c(s) = k' \left( \frac{\lambda s + 1}{x\lambda s + 1} \right)^{\alpha} \tag{23}$$

For $k' = \alpha = 1$, $\lambda = \frac{K_p}{K_i}$, and $x$ has a very high value (for example $x > 10,000$), the transfer function of the FO-lead-lag controller becomes the transfer function of the FO-PI controller. It can therefore be concluded that there is an increased flexibility in the use of the FO-lead-lag controller in a control loop.

### 4.4. FO-SMC Speed Controller

To achieve an SMC-type controller for the control of a PMSM motor described by Equation (14), the state variables $x_1$ and $x_2$ described below are selected [1]:

$$x_1 = \omega_{ref} - \omega \tag{24}$$

where $x_1$ represents the tracking error of the speed.

$$x_2 = \dot{x}_1 = \frac{\omega_{ref} - \omega}{dt} = -\dot{\omega} \tag{25}$$

Equation (26) defines the sliding surface $S$ of the zero-error manifold. Through differentiation, the following equation is obtained (27):

$$S = cx_1 + x_2 \tag{26}$$

$$\dot{S} = cx_2 + \dot{x}_2 = cx_2 - D\dot{i}_q \tag{27}$$

where $c$ represents the positive adjustable parameter and $D = \frac{3n_p \lambda_0}{2J}$ is obtained from Equation (13).

In order to control the response time of the PMSM control system, the condition of time evolution of the surface $S$ is imposed like in Equation (28):

$$\dot{S} = -\varepsilon \text{sgn}(S) - qS, \quad \varepsilon, q > 0 \tag{28}$$

where $\varepsilon$ and $q$ represent the positive adjustable parameters; *sgn()* represents the *signum* function.

To reduce the chattering effect (characteristic of the SMC design), the *sgn* function is replaced with the sigmoid function defined as follows:

$$H(x) = \frac{2}{1 + e^{-a(x-c)}} - 1 \tag{29}$$

For $a = 4$ and $c = 0$, $H \in [-1\,1]$, the transition of the function $H$ from $-1$ to $1$ is smoothed and ensures the reduction of the chattering effect. Based on these, after some calculations, the SMC-type controller output value is obtained in the following form:

$$i_{qref}(t) = \frac{1}{D} \int_0^t [cx_2 + \varepsilon H(S) + qS] dt \tag{30}$$

It can be specified that $i_{qref}$ represents the current reference for the control loop on q axis, while $i_{dref}$ is set to zero according to the FOC-type control strategy [23]. To demonstrate the stability of the PMSM control system under the action of the control law given by Equation (30), the Lyapunov function candidate is selected in the following form $V = \frac{1}{2}S^2$ [1].

$$\dot{V} = S\dot{S} = S[-\varepsilon H(S) - qS] = -\varepsilon H(S) - qS^2 \tag{31}$$

After some calculations, $\dot{V} \leq 0$ is obtained, where $\dot{V}$ is given by the relation (31). To achieve the FO-SMC controllers, the sliding surface $S$ is selected as follows:

$$S = k_p x_1 + k_d D^\mu x_1 = k_p x_1 + k_d D^{\mu-1} x_2 \tag{32}$$

After differentiation, the following relation is obtained:

$$\dot{S} = k_p \dot{x}_1 + k_d D^{\mu+1} x_1 = k_p x_2 + k_d D^{\mu-1} \dot{x}_2 \tag{33}$$

Based on the mathematical model of the PMSM described in Section 3, the following relation is obtained:

$$\dot{x}_2 = \ddot{\omega}_{ref} - \frac{3n_p\lambda_0}{2J}\dot{i}_q + \frac{1}{J}\dot{T}_L + \frac{B}{J}\dot{\omega} \tag{34}$$

By inserting Equation (34) into Equation (33), the following relation is obtained:

$$\dot{S} = k_p x_2 + k_d D^{\mu-1}\left(\ddot{\omega}_{ref} - \frac{3n_p\lambda_0}{2J}\dot{i}_q + \frac{1}{J}\dot{T}_L + \frac{B}{J}\dot{\omega}\right) \tag{35}$$

For $\dot{S} = 0$, the following relation is obtained:

$$-\varepsilon H(S) - qS - k_p x_2 = k_d D^{\mu-1}\left(\ddot{\omega}_{ref} - \frac{3n_p\lambda_0}{2J}\dot{i}_q + \frac{1}{J}\dot{T}_L + \frac{B}{J}\dot{\omega}\right) \tag{36}$$

By applying operator $D^{1-\mu}$ (described in Section 2) to both members of Equation (36), the following relation is obtained:

$$D^{1-\mu}\left(-\varepsilon H(S) - qS - k_p x_2\right) = k_d\left(\ddot{\omega}_{ref} - \frac{3n_p\lambda_0}{2J}\dot{i}_q + \frac{1}{J}\dot{T}_L + \frac{B}{J}\dot{\omega}\right) \tag{37}$$

This results in the following relation:

$$\frac{1}{k_d}D^{1-\mu}\left(-\varepsilon H(S) - qS - k_p x_2\right) = \ddot{\omega}_{ref} - \frac{3n_p\lambda_0}{2J}\dot{i}_q + \frac{1}{J}\dot{T}_L + \frac{B}{J}\dot{\omega} \tag{38}$$

Equation (38) can be rewritten as follows:

$$\frac{1}{J}\frac{3}{2}n_p\lambda_0\dot{i}_q = \ddot{\omega}_{ref} + \frac{1}{J}\dot{T}_L + \frac{B}{J}\dot{\omega} - \frac{1}{k_d}D^{1-\mu}\left(-\varepsilon H(S) - qS - k_p x_2\right) \tag{39}$$

The current reference $i_{qref}$ is obtained from Equation (39) as follows:

$$i_{qref}(t) = \frac{1}{\frac{1}{J}\frac{3}{2}n_p\lambda_0}\int_0^t\left[\ddot{\omega}_{ref} + \frac{1}{J}\dot{T}_L + \frac{B}{J}\dot{\omega} - \frac{1}{k_d}D^{1-\mu}\left(-\varepsilon H(S) - qS - k_p x_2\right)\right]dt \tag{40}$$

## 5. Fractional Order Synergetic Current Controllers for PMSM

The synergetic control can be considered as a generalization of the sliding mode control (SMC), retaining the decoupling design procedure and model order reduction properties of this type of control and the advantages provided by this approach for the synthesis of the control. Thus, in this section, the PI-type current controllers will be replaced for the inner current control loops in the FOC-type control strategy, with synergetic and FO-synergetic type controllers to obtain superior performances.

For the synergetic control, a macro-variable is defined as a function of the states of the system, as follows [19]:

$$\Psi = \Psi(x, t) \tag{41}$$

The synthesized control inputs will force the system to operate on the manifold $\Psi = 0$, in a similar manner to the SMC. A number of macro-variables equal to the number of control inputs are defined. The dynamic evolution of each macro-variable is defined according to the following equation:

$$T\dot{\Psi} + \Psi = 0, \, T > 0 \tag{42}$$

where $T$ is selected so as to achieve the rate of convergence of the system evolution towards the desired manifold.

By differentiating the macro-variable $\Psi$:

$$\dot{\Psi} = \frac{d\Psi}{dx}\dot{x}, \tag{43}$$

and by inserting Equation (43) into (42) using the explicit description of the states $\dot{x}$ from the mathematical model, in the case of the PMSM expressed by Equation (14), the control law is obtained as follows:

$$u = u(x, \Psi(x, t), T, t) \tag{44}$$

In case of applying the FOC type strategy for the PMSM control (see Figure 4), the outer speed control loop supplies the reference $i_{qref}$ at the PI-type speed controller output for the inner control loop whose controller proposed in this paper is synergetic. According to Equations (42)–(44), the synergetic controller provides the controls $u_d$ and $u_q$. Furthermore, according to the FOC control strategy of the PMSM $i_{dref} = 0$.

According to [20], in order to achieve superior control performance under static and dynamic regime, $\omega_{acc}$ and $\omega_{dec}$ are defined as the angular velocity of the rotor by selecting the current limit mode of operation for accelerating and decelerating transients, respectively, of the following form:

$$\begin{aligned}\omega_{acc} &= \omega_{ref} - k_q\left(\left|i_{qmax}\right| - i_{qref}\right)\\ \omega_{dec} &= \omega_{ref} - k_q\left(-\left|i_{qmax}\right| - i_{qref}\right)\end{aligned} \tag{45}$$

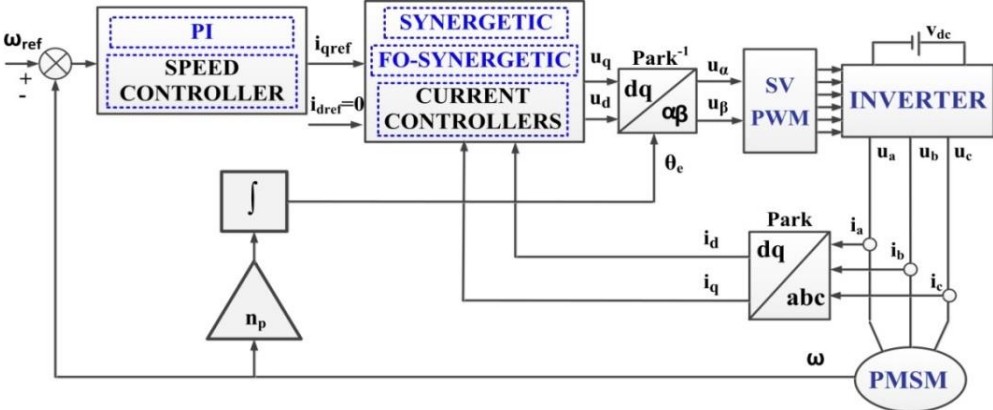

**Figure 4.** FO-synergetic current control for the PMSM—general block diagram.

For $\omega > \omega_{acc}$ and $\omega < \omega_{dec}$ the macro-variable on q axis $\Psi_q$ can be represented as in the following equation:

$$\Psi_q = \left(\omega(t) - \omega_{ref}\right) + k_q\left(i_q(t) - i_{qref}\right) \tag{46}$$

and for $\omega \leq \omega_{acc}$ the macro-variable on q axis $\Psi_q$ is represented in Equation (47), and for $\omega \geq \omega_{dec}$ the macro-variable on q axis $\Psi_q$ is represented in Equation (48).

$$\Psi_q = i_q(t) - \left|i_{qmax}\right| + k_{iq}\int_0^t \left(i_q(t) - \left|i_{qmax}\right|\right)dt \tag{47}$$

$$\Psi_q = i_q(t) + \left|i_{qmax}\right| + k_{iq}\int_0^t \left(i_q(t) + \left|i_{qmax}\right|\right)dt \tag{48}$$

where $i_{qmax}$ is the maximum admissible current on q-axis, and $k_q$ is a value which is dynamically adjusted as a function of the angular velocity of the rotor error. The control law for q axis can be expressed as follows: for $\omega_{acc} < \omega < \omega_{dec}$, $u_q$ is given by Equation (49), for $\omega \leq \omega_{acc}$, $u_q$ is given by Equation (50), and for $\omega \geq \omega_{dec}$, $u_q$ is given by Equation (51):

$$u_q(t) = R_s i_q + n_p\omega(Li_d + \lambda_0) + \frac{L}{T_q}\left(i_{qref} - i_q\right) + \frac{L}{T_q k_q}\left(\omega_{ref} - \omega\right) + \frac{L}{Jk_q}\left(-K_t i_q + B\omega + T_L\right) \tag{49}$$

$$u_q(t) = R_s i_q + n_p\omega(Li_d + \lambda_0) + \frac{L}{T_q}\left(\left|i_{qmax}\right| - i_q\right) + k_{iq}L\left(\left|i_{qmax}\right| - i_q\right) - \frac{k_{iq}L}{T_q}\int_0^t \left(i_q - \left|i_{qmax}\right|\right)dt \tag{50}$$

$$u_q(t) = R_s i_q + n_p\omega(Li_d + \lambda_0) + \frac{L}{T_q}\left(-\left|i_{qmax}\right| - i_q\right) + k_{iq}L\left(-\left|i_{qmax}\right| - i_q\right) - \frac{k_{iq}L}{T_q}\int_0^t \left(i_q + \left|i_{qmax}\right|\right)dt \tag{51}$$

The macro-variable on d axis $\Psi_d$ can be represented as in the following equation:

$$\Psi_d = \left(i_d(t) - i_{dref}\right) + k_{id}\int_0^t \left(i_d(t) - i_{dref}\right)dt \tag{52}$$

After some calculus, the control law for d axis can be expressed as follows [20]:

$$u_d(t) = R_s i_d - n_p \omega L i_q - \frac{L}{T_d}\left(i_d - i_{dref}\right) - k_{id}L\left(i_d - i_{dref}\right) - \frac{k_{id}L}{T_d}\int_0^t \left(i_d - i_{dref}\right)dt \tag{53}$$

Following the calculations made for the synergetic control of the PMSM, using definitions (1) and (2) of the fractional calculus for integration and differentiation operators, in this section we will obtain the values of the controls $u_d(t)$ and $u_q(t)$.

We define the macro-variable on q axis $\Psi_q$ similarly to (46) for $\omega > \omega_{acc}$ and $\omega < \omega_{dec}$, as follows:

$$\Psi_q(x) = D^\mu x_3 + k_q x_1 \tag{54}$$

where: $x_1 = i_q - i_{qref}$, $x_3 = \omega - \omega_{ref}$, and $\mu > 0$. Next, $\dot{\Psi}_q$ is calculated, and the following relation is obtained:

$$\dot{\Psi}_q(x) = D^\mu \dot{x}_3 + k_q \dot{x}_1 = D^\mu \dot{\omega} + k_q \dot{i}_q \tag{55}$$

By inserting (55) into the Equation (42) of dynamic evolution of macro-variable $\Psi_q$, for $T = T_q$, the following relation is obtained:

$$T_q\left[D^\mu\left(\frac{K_t i_q}{J} - \frac{B\omega}{J} - \frac{T_L}{J}\right) + k_q\left(-\frac{R_s i_q}{L} - n_p \omega i_d - \frac{n_p \lambda_0 \omega}{L} + \frac{u_q}{L}\right)\right] + \\ + D^\mu\left(\omega - \omega_{ref}\right) + k_q\left(i_q - i_{qref}\right) = 0 \tag{56}$$

By rearranging the terms in Equation (56) the following relation is obtained:

$$T_q D^\mu\left(\frac{K_t i_q}{J} - \frac{B\omega}{J} - \frac{T_L}{J}\right) + T_q k_q\left(-\frac{R_s i_q}{L} - n_p \omega i_d - \frac{n_p \lambda_0 \omega}{L}\right) + \\ + \frac{T_q k_q u_q}{L} + D^\mu\left(\omega - \omega_{ref}\right) + k_q\left(i_q - i_{qref}\right) = 0 \tag{57}$$

After some calculations, Equation (56) becomes:

$$\frac{T_q k_q u_q}{L} = T_q D^\mu\left(-\frac{K_t i_q}{J} + \frac{B\omega}{J} + \frac{T_L}{J}\right) + \frac{T_q k_q R_s i_q}{L} + \\ + n_p \omega(L i_d + \lambda_0)\frac{T_q k_q}{L} + D^\mu\left(\omega - \omega_{ref}\right) + k_q\left(i_q - i_{qref}\right) = 0 \tag{58}$$

Based on this, the control $u_q$ of the PMSM is obtained:

$$u_q(t) = \frac{L}{J k_q}D^\mu\left(-K_t i_q + B\omega + T_L\right) + R_s i_q + n_p \omega(L i_d + \lambda_0) \\ + \frac{L}{T_q k_q}D^\mu\left(\omega - \omega_{ref}\right) + \frac{L}{T_q}\left(i_q - i_{qref}\right) \tag{59}$$

Similarly to relations (47) and (48), control $u_q$ is obtained for $\omega \le \omega_{acc}$ and $\omega \ge \omega_{dec}$, respectively. Next, the macro-variable on d axis $\Psi_d$ is defined:

$$\Psi_d(x,t) = D^\mu x_2 + k_d \int_0^t x_2(t)dt \tag{60}$$

where $x_2 = i_d - i_{dref}$ and $\mu > 0$.

Next, $\dot{\Psi}_d$ is calculated, and the following relation is obtained:

$$\dot{\Psi}_d(x) = D^\mu \dot{x}_2 + k_d x_2 = D^\mu \dot{i}_d + k_d\left(i_d - i_{dref}\right) \tag{61}$$

By inserting (61) into the Equation (42) of dynamic evolution of macro-variable $\Psi_d$, for $T = T_d$, the following relation is obtained:

$$T_d\left\{\left[D^\mu\left(n_p\omega i_q - \frac{R_s i_d}{L} + \frac{u_d}{L}\right)\right] + k_d\left(i_d - i_{dref}\right)\right\} + D^\mu\left(i_d - i_{dref}\right) + k_d\int_0^t\left(i_d - i_{dref}\right)dt = 0 \qquad (62)$$

By applying the operator defined in (1), $D^{-\mu}$ (which becomes $I_\mu$) to both members of Equation (62) the following relation is obtained:

$$T_d\left(n_p\omega i_q - \frac{R_s i_d}{L} + \frac{u_d}{L}\right) + T_d k_d I_\mu\left(i_d - i_{dref}\right) + i_d - i_{dref} + k_d I_{\mu+1}\left(i_d - i_{dref}\right) = 0 \qquad (63)$$

After some calculations, Equation (63) becomes:

$$\frac{T_d u_d}{L} = \frac{R_s i_d T_d}{L} - n_p\omega i_q T_d - T_d k_d I_\mu\left(i_d - i_{dref}\right) - \left(i_d - i_{dref}\right) - k_d I_{\mu+1}\left(i_d - i_{dref}\right) = 0 \qquad (64)$$

Based on this, the control $u_d$ of the PMSM is obtained:

$$u_d(t) = R_s i_d - n_p\omega i_q L - L k_d I_\mu\left(i_d - i_{dref}\right) - \frac{L}{T_d}\left(i_d - i_{dref}\right) - \frac{L k_d}{T_d}I_{\mu+1}\left(i_d - i_{dref}\right) \qquad (65)$$

## 6. Rotor Speed Estimation and Fault Detection

This section presents two observers, which complete the classic FOC-type control structure. Thus, the sensorless characteristic of the control is ensured by an SMO-type observer which estimates the PMSM speed. Furthermore, the use of an FDO-type observer enables the fault detection of the current sensors on the PMSM supply lines.

### 6.1. Rotor Speed and Position Estimations Based on SMO-Type Observer

By using the PMSM operating equations given by the relations (11)–(14), and by applying the inverse Park transform, the equations of the currents $i_\alpha$ and $i_\beta$ and the back-EMF $e_\alpha$ and $e_\beta$ are obtained in α-β frame [2]:

$$\begin{aligned} i_\alpha &= i_d\cos(\theta_e) - i_q\sin(\theta_e) \\ i_\beta &= i_d\sin(\theta_e) + i_q\cos(\theta_e) \end{aligned} \qquad (66)$$

$$\begin{aligned} e_\alpha &= \frac{d\lambda_\alpha}{dt} = -\lambda_0\omega_e\sin(\theta_e) \\ e_\beta &= \frac{d\lambda_\beta}{dt} = -\lambda_0\omega_e\cos(\theta_e) \end{aligned} \qquad (67)$$

Based on these, the PMSM operating equations can be rewritten as:

$$\begin{aligned} \frac{di_\alpha}{dt} &= -\frac{R_s}{L}i_\alpha - \frac{1}{L}e_\alpha + \frac{1}{L}u_\alpha \\ \frac{di_\beta}{dt} &= -\frac{R_s}{L}i_\beta - \frac{1}{L}e_\beta + \frac{1}{L}u_\beta \end{aligned} \qquad (68)$$

The equations of the SMO-type observer according to which the rotor speed and position can be estimated are given by the equations [23,25]:

$$\begin{aligned} \frac{d\hat{i}_\alpha}{dt} &= -\frac{R_s}{L}\hat{i}_\alpha + \frac{1}{L}u_\alpha - \frac{1}{L}kH(\hat{i}_\alpha - i_\alpha) \\ \frac{d\hat{i}_\beta}{dt} &= -\frac{R_s}{L}\hat{i}_\beta + \frac{1}{L}u_\beta - \frac{1}{L}kH(\hat{i}_\beta - i_\beta) \end{aligned} \qquad (69)$$

where: $k$ represents the observer gain, and $H$ is a sigmoid type function described in Equation (29).

The sliding vector is selected as follows:

$$S_n = \left[S_\alpha\ S_\beta\right]^T = \left[\hat{i}_\alpha - i_\alpha\ \hat{i}_\beta - i_\beta\right]^T = \left[\bar{i}_\alpha\ \bar{i}_\beta\right]^T \qquad (70)$$

To demonstrate the stability of the proposed observer, a Lyapunov function is selected of the form [25]:

$$V = \frac{1}{2}S_n^T S_n = \frac{1}{2}\left(S_\alpha^2 + S_\beta^2\right) \tag{71}$$

The current error system is defined in the form:

$$\begin{aligned}
\dot{\bar{i}}_\alpha = \dot{\hat{i}}_\alpha - \dot{i}_\alpha = -\frac{R_s}{L}\bar{i}_\alpha + \frac{1}{L}e_\alpha - \frac{1}{L}kH(\bar{i}_\alpha) \\
\dot{\bar{i}}_\beta = \dot{\hat{i}}_\beta - \dot{i}_\beta = -\frac{R_s}{L}\bar{i}_\beta + \frac{1}{L}e_\beta - \frac{1}{L}kH(\bar{i}_\beta)
\end{aligned} \tag{72}$$

According to these, $\dot{V}$ is calculated, and the following relation is obtained:

$$\dot{V} = -\frac{R_s}{L}\left(\bar{i}_\alpha^2 + \bar{i}_\beta^2\right) + \frac{1}{L}\left[(e_\alpha - k)\bar{i}_\alpha H(\bar{i}_\alpha) + \left(e_\beta - k\right)\bar{i}_\beta H(\bar{i}_\beta)\right] < 0 \tag{73}$$

By selecting the observer gain as $k \geq \max\left(|e_\alpha|, |e_\beta|\right)$, the stability condition of the observer is obtained: $\dot{V} < 0$.

Based on this, on the sliding surface, the following relation is obtained:

$$[\dot{S}_\alpha \ \dot{S}_\beta]^T = [S_\alpha \ S_\beta]^T \approx [0 \ 0] \tag{74}$$

Based on relations (73) and (74), are obtained the estimations for $e_\alpha$ and $e_\beta$:

$$\begin{aligned}
\hat{e}_\alpha = kH(\bar{i}_\alpha) = -\lambda_0\hat{\omega}_e \sin\theta_e \\
\hat{e}_\beta = kH(\bar{i}_\beta) = \lambda_0\hat{\omega}_e \cos\theta_e
\end{aligned} \tag{75}$$

Finally, the estimates for the speed and position of the PMSM rotor can be obtained as below:

$$\begin{cases}
\hat{\omega}_e = \dfrac{\sqrt{\hat{e}_\alpha^2 + \hat{e}_\beta^2}}{\lambda_0} \\
\hat{\theta}_e(t) = \displaystyle\int_{t_0}^{t} \hat{\omega}_e(t)dt + \theta_0
\end{cases} \tag{76}$$

where: $\theta_0$ represents the initial electrical position of the rotor.

Figure 5 shows the implementation in the MATLAB/Simulink environment in order to perform the numerical simulations of the SMO-type observer for the estimation of the position and rotor speed of the PMSM.

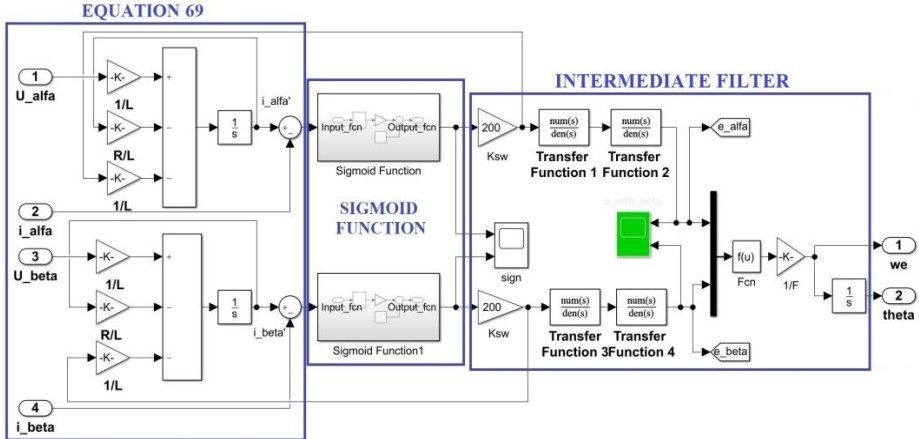

**Figure 5.** Sliding mode observer (SMO)-type observer—MATLAB/Simulink implementation block diagram.

## 6.2. Fault Detection Based on FDO-Type Observer

To detect the faults of the current sensors on the PMSM supply phases, an FDO-type observer will be used. In the α-β frame, the equations analogous to those described in Equation (11) for a PMSM are as follows:

$$
\begin{bmatrix} u_\alpha \\ u_\beta \end{bmatrix} = R_s \begin{bmatrix} i_\alpha \\ i_\beta \end{bmatrix} + \rho L \begin{bmatrix} i_\alpha \\ i_\beta \end{bmatrix} + \omega_e \begin{bmatrix} \lambda_0 & 0 \\ 0 & \lambda_0 \end{bmatrix} \begin{bmatrix} -\sin\theta_e \\ \cos\theta_e \end{bmatrix}
\tag{77}
$$

According to Equation (77), the following relation can be written:

$$
\rho \begin{bmatrix} i_\alpha \\ i_\beta \end{bmatrix} = \frac{R_s}{L} \begin{bmatrix} i_\alpha \\ i_\beta \end{bmatrix} + \frac{1}{L} \begin{bmatrix} u_\alpha \\ u_\beta \end{bmatrix} + \frac{\omega_e}{L} \begin{bmatrix} \lambda_0 & 0 \\ 0 & \lambda_0 \end{bmatrix} \begin{bmatrix} -\sin\theta_e \\ \cos\theta_e \end{bmatrix}
\tag{78}
$$

Considering the flux linkage $\lambda_{ext} = \lambda_0$ in α-β frame is obtained the following relation:

$$
\lambda_{ext,\alpha\beta} = \begin{bmatrix} \lambda_{ext,\alpha} \\ \lambda_{ext,\beta} \end{bmatrix} = \begin{bmatrix} \lambda_0 & 0 \\ 0 & \lambda_0 \end{bmatrix} \begin{bmatrix} \cos\theta \\ \sin\theta \end{bmatrix}
\tag{79}
$$

Let us note: $x = \begin{bmatrix} i_\alpha & i_\beta \end{bmatrix}^T$, $u = \begin{bmatrix} u_\alpha & u_\beta \end{bmatrix}^T$, $A = \begin{bmatrix} -\frac{R_s}{L} & 0 \\ 0 & -\frac{R_s}{L} \end{bmatrix}$, $B = \begin{bmatrix} \frac{1}{L} & 0 \\ 0 & \frac{1}{L} \end{bmatrix}$, $C = \begin{bmatrix} 1 & 0 \\ 0 & 1 \end{bmatrix}$, $F = \begin{bmatrix} 0 & \frac{\omega_e}{L} \\ \frac{\omega_e}{L} & 0 \end{bmatrix} = -\frac{\omega_e}{L}J$, $J = \begin{bmatrix} 0 & -1 \\ 1 & 0 \end{bmatrix}$.

According to Equation (78), the equation of the PMSM which has an embedded FDO-type observer can be written as [40]:

$$
\begin{cases} \dot{x}(t) = Ax(t) + Bu(t) + F\lambda_{ext,\alpha\beta} + Ed \\ \quad\quad y(t) = Cx(t) + Gf_s \end{cases}
\tag{80}
$$

where $f_s = \begin{bmatrix} f_{s\alpha} & f_{s\beta} \end{bmatrix}^T$ is the stator current sensor fault vector, $d = \begin{bmatrix} d_1 & d_2 \end{bmatrix}^T$ is the unknown but bounded disturbance vector, $x$ is the state vector, $u$ and $y$ represents the input and output vector, respectively, $G = \begin{bmatrix} 1 & 0 \\ 0 & 1 \end{bmatrix}$, and $E = \begin{bmatrix} 1 & 0 \\ 0 & 1 \end{bmatrix}$.

Consider a new state variable $z$, which is just variable $y$, but filtered, with $a$, $b$ constants:

$$
\dot{z} = -az + by
\tag{81}
$$

By selecting $a = 0$, $b = 1$, Equation (81) becomes:

$$
\dot{z} = y = Cx(t) + Gf_s
\tag{82}
$$

Based on relations (80)–(82), the system of equations of the PMSM with embedded FDO observer becomes:

$$
\begin{cases} \dot{x}(t) = Ax(t) + Bu(t) + F\lambda_{ext,\alpha\beta} + Ed \\ \dot{z}(t) = Cx(t) + Gf_s \\ \quad w = z \end{cases}
\tag{83}
$$

Consider $f_s$ the actuator fault of the system described by (83). The FDO-type observer has the following form [40]:

$$
\begin{cases} \dot{\hat{x}}(t) = A\hat{x}(t) + Bu(t) + F\lambda_{ext,\alpha\beta} + Ev_1 \\ \dot{\hat{z}}(t) = C\hat{x}(t) + G\hat{f}_s + v_2 \end{cases}
\tag{84}
$$

$v_1$ and $v_2$ are selected, as control signals for the correction of the sliding mode, as follows:

$$
\begin{cases} v_1 = L_1 H(e_x) \\ v_2 = L_2 H(e_z) \end{cases}
\tag{85}
$$

where $e_x = x - \hat{x}$ and $e_z = z - \hat{z}$, and $L_1$ and $L_2$ are positive design constants. Let us note $e_s = f_s - \hat{f}_s$.

Based on this, the equations of the errors can be written as follows:

$$\begin{cases} \dot{e}_x = \dot{x} - \dot{\hat{x}} = Ae_x + E(d - v_1) \\ \dot{e}_z = \dot{z} - \dot{\hat{z}} = Ce_x + Ge_s - v_2 \end{cases} \tag{86}$$

To demonstrate the stability of FDO-type observers, the Lyapunov function is selected as follows:

$$\dot{V} = e_x^T e_x + e_z^T e_z + e_s^T Q e_s \tag{87}$$

where $Q > 0$ is a constant matrix with appropriate dimensions.

By calculating $\dot{V}$, the following relation is obtained:

$$\dot{V} \leq 2\|e_x\| \ \|E\|(\|d\| - L_1) + 2\|e_z\|(\|C\| \ \|e_x\| - L_2) + 2e_s^T\left(Ge_z - Q\dot{\hat{f}}_s\right) \tag{88}$$

To ensure stability, by selecting $L_1 > \|d\|$ and $L_2 > C\|e_x\|$, the law of adaptation for faults vector $f_s$ is obtained:

$$\dot{\hat{f}}_s = Q^{-1} Ge_z \tag{89}$$

Since $i_\alpha$ and $i_\beta$ can be derived from $i_{a,b,c}$ in the form:

$$\begin{cases} i_\alpha = i_a \\ i_\beta = \frac{i_b - i_c}{\sqrt{3}} = \frac{2i_b + i_a}{\sqrt{3}} \end{cases}, \tag{90}$$

Then, the effect of the occurrence of faults $f_{s\alpha}$ and $f_{s\beta}$, propagate in the form of faults $f_a$ and $f_b$ in phases a and b of the PMSM power windings in the form of equations and laws of adaptation described by Equations (91)–(93):

$$\begin{cases} f_{s\alpha} = f_a \\ f_{s\beta} = \frac{2f_b + f_a}{\sqrt{3}} \end{cases} \tag{91}$$

$$\dot{\hat{f}}_a = \dot{\hat{f}}_{s\alpha} = Q^{-1} e_{z1} \tag{92}$$

$$\dot{\hat{f}}_b = \frac{\sqrt{3}\dot{\hat{f}}_{s\beta} - \dot{\hat{f}}_{s\alpha}}{2} = Q^{-1}\frac{\sqrt{3}e_{z2} - e_{z1}}{2} \tag{93}$$

In order to achieve the implementation of the FDO-type observer in the MATLAB/Simulink environment, it is necessary to define the equations presented in this section explicitly, by components. For implementation, $Q = G = I_2$. Thus, Equation (83) is defined explicitly in the form of Equations (94)–(96).

$$\begin{cases} \dot{i}_\alpha = -\frac{R_s}{L}i_\alpha + \frac{u_\alpha}{L} + \frac{\omega_e \lambda_0}{L} + d_1 \\ \dot{i}_\beta = -\frac{R_s}{L}i_\beta + \frac{u_\beta}{L} - \frac{\omega_e \lambda_0}{L} + d_2 \end{cases} \tag{94}$$

$$\begin{cases} \dot{z}_\alpha = y_\alpha = i_\alpha + f_\alpha \\ \dot{z}_\beta = y_\beta = i_\beta + f_\beta \end{cases} \tag{95}$$

$$\begin{cases} z_\alpha = \frac{1}{s}(i_\alpha + f_\alpha) \\ z_\beta = \frac{1}{s}(i_\beta + f_\beta) \end{cases} \tag{96}$$

Equations (84) and (85) are defined explicitly by components in the form of Equations (97)–(99).

$$\begin{cases} \dot{\hat{i}}_\alpha = -\frac{R_s}{L}\hat{i}_\alpha + \frac{u_\alpha}{L} + \frac{\omega_e \lambda_0}{L} + L_1 H\left(i_\alpha - \hat{i}_\alpha\right) \\ \dot{\hat{i}}_\beta = -\frac{R_s}{L}\hat{i}_\beta + \frac{u_\beta}{L} - \frac{\omega_e \lambda_0}{L} + L_1 H\left(i_\beta - \hat{i}_\beta\right) \end{cases} \tag{97}$$

$$\begin{cases} \dot{\hat{z}}_\alpha = \hat{i}_\alpha + \hat{f}_\alpha + L_2 H(z_\alpha - \hat{z}_\alpha) \\ \dot{\hat{z}}_\beta = \hat{i}_\beta + \hat{f}_\beta + L_2 H(z_\beta - \hat{z}_\beta) \end{cases} \tag{98}$$

$$\begin{cases} e_{z\alpha} = z_\alpha - \hat{z}_\alpha \\ e_{z\beta} = z_\beta - \hat{z}_\beta \end{cases} \tag{99}$$

Equation (89) is implemented as follows:

$$\begin{bmatrix} \hat{f}_\alpha \\ \hat{f}_\beta \end{bmatrix} = \begin{bmatrix} z_\alpha - \hat{z}_\alpha \\ z_\beta - \hat{z}_\beta \end{bmatrix}; \quad \begin{cases} \hat{f}_\alpha = \frac{1}{s}(z_\alpha - \hat{z}_\alpha) \\ \hat{f}_\beta = \frac{1}{s}(z_\beta - \hat{z}_\beta) \end{cases} \tag{100}$$

We specify that s represents the complex variable in Equations (70)–(73). Figure 6 shows the implementation in MATLAB/Simulink of the FDO-type observer using the equations defined explicitly by components (94)–(100).

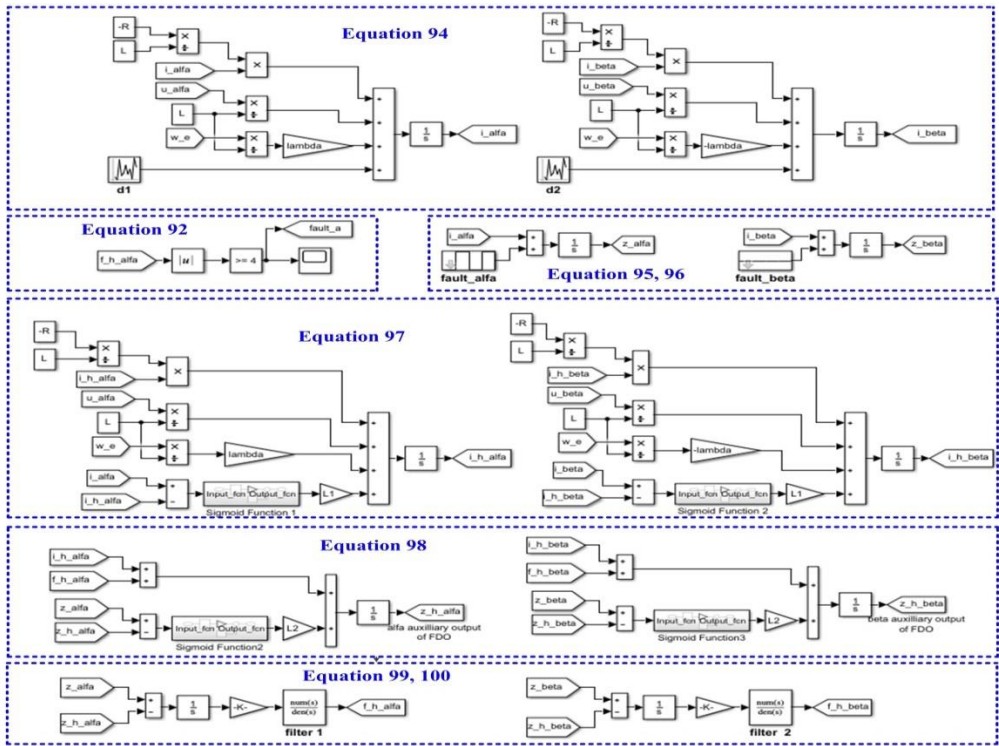

**Figure 6.** FDO-type observer—MATLAB/Simulink implementation block diagram.

## 7. Numerical Simulations

For the numerical simulations performed in MATLAB/Simulink and presented in this section, the PMSM parameters are shown in Table 1.

**Table 1.** Nominal parameters of PMSM.

| Motor Parameter | Symbol | Value | Unit |
|---|---|---|---|
| Stator resistance | $R_s$ | 2.875 | $\Omega$ |
| d axes inductance | $L_d$ | 0.0085 | H |
| q axes inductance | $L_q$ | 0.0085 | H |
| Combined inertia of rotor and load | $J$ | 0.0008 | kg·m$^2$ |
| Combined viscous friction of rotor and load | $B$ | 0.005 | N·m·s/rad |
| Flux induced by permanent magnets of rotor in stator phases | $\lambda_0$ | 0.175 | Wb |
| Pole pairs number | $n_p$ | 4 | - |

*7.1. Numerical Simulations—Fractional Order Speed Controllers for PMSM*

The block diagram for the implementation in MATLAB/Simulink of the PMSM control system based on the FOC strategy proposed in this article is presented in Figure 7. The results of the behavior of the PMSM control system are presented comparatively, using the classic PI controller, FO-PI controller, TID controller, FO-lead-lag controller and FO-SMC controller for the control of the outer PMSM rotor speed control loop. The other blocks in Figure 7 implement the speed reference generator, the load torque generator, the time evolution I/O signals and the model of the PMSM drive. The sensorless function of the control system of the rotor speed is provided by the use of an SMO-type observer described in Section 6. Additionally, to detect the faults of the current sensors on the supply phases of the PMSM, an FDO-type observer as described in Section 6 is embedded in the general control structure in Figure 7.

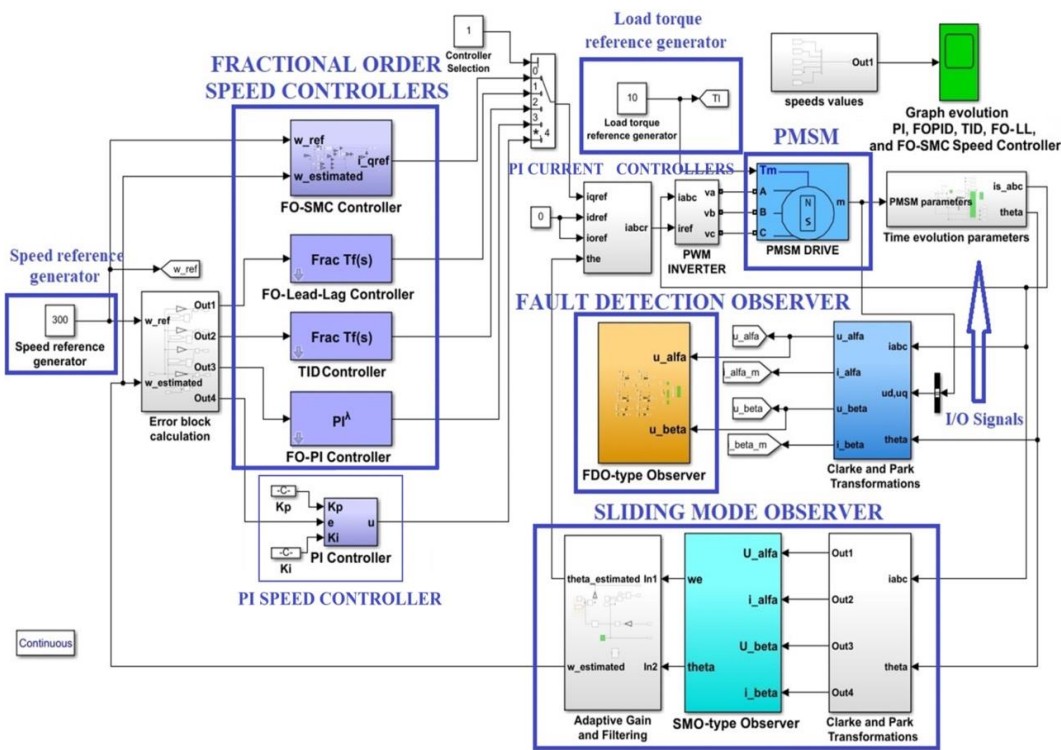

**Figure 7.** MATLAB/Simulink implementation block diagram for sensorless control of PMSM based on fractional order speed controllers, PI current controllers, SMO speed observer and FDO.

Starting from Equations (17) and (18), which represent the transfer function of the fixed part of the PMSM rotor speed control system, for the nominal parameters presented in Table 1, the theoretical transfer function of the fixed part is obtained:

$$H_f(s) = \frac{2.857}{5.44 \cdot 10^{-4} s^3 + 2.588 \cdot 10^{-5} s^2 + 0.03761\, s + 0.7637} \tag{101}$$

The discrete form obtained for the transfer function based on Equation (101) according to the Tustin method for a sampling period of 0.1 ms is:

$$H_f(z) = \frac{0.004648 z^3 + 0.01394 z^2 + 0.01394 z + 0.004648}{z^3 - 2.166 z^2 + 1.837 z - 0.6607} \tag{102}$$

The transfer function of the fixed part of the PMSM rotor speed control system is obtained by using the MATLAB System Identification Toolbox, through identification in continuous and discrete form in Equations (103) and (104), respectively:

$$H_{f\_ident}(s) = \frac{0.002373}{s^3 + 0.0663s^2 + 0.01484s + 8.163 \cdot 10^{-6}} \tag{103}$$

$$H_{f\_ident}(z) = \frac{2.96 \cdot 10^{-3} z^3 + 8.89 \cdot 10^{-3} z^2 + 8.89 \cdot 10^{-3} z + 2.96 \cdot 10^{-3}}{z^3 - 3z^2 + 3z - 0.99} \tag{104}$$

The tuning of PI controllers by using Ziegler–Nichols methods is a well-known technique. In the fractional case, the FOMCON toolbox for the MATLAB utility program is used for the tuning of FO-PI controllers. In order to obtain the optimal tuning parameters in the fractional case, a number of optimization methods are incorporated in the FOMCON toolbox, both in the frequency range and in the time domain. In the frequency range, the goal of optimizing the parameters is achieved by obtaining optimal performance in terms of the sensitivity function $S(j\omega)$ for disturbance rejection for the low and middle frequency range and the rejection of the high frequency noise using the complementary sensitivity function $T(j\omega)$. In the time domain, the tuning of fractional controllers is carried out by minimizing optimal criteria, such as the integral absolute error (IAE) [41].

Using the FOMCON toolbox for MATLAB, an FO-PI controller described by Equation (20) can be tuned for the control of the PMSM rotor speed. For $K_p = 1.2$, $K_i = 12$, $\lambda = 1.1$, and $K_d = \mu = 0$, the transfer function of the FO-PI controller is obtained:

$$H_{PI} = \frac{1.2s^{1.1} + 12}{s^{1.1}} \tag{105}$$

The closed loop transfer function of the PMSM rotor speed control, where the controller is given by Equation (20) and the fixed part is given by Equation (103), is expressed in the following form:

$$H_{CL\_PI} = \frac{4.3272 \cdot 10^8 s^{1.1} + 4.3272 \cdot 10^9}{s^{3.2} + 73730s^{2.2} + 4.3746 \cdot 10^8 s^{1.1} + 4.3272 \cdot 10^9} \tag{106}$$

Using the FOMCON toolbox for MATLAB, the stability of the closed loop system of the PMSM rotor speed control, in other words, the fulfillment of the condition given by the relation (6) is graphically presented in Figure 8, where it is noted that the system is stable.

The step response for the FO-PI controller given by the relation (105), considering the fixed part is represented by both the theoretical transfer function and by the transfer function obtained through identification, in other words, the relations (101) and (103), respectively, is presented in Figure 9.

A similar response is noted in the two cases, with a good behavior under both the dynamic and stationary regimes and which additionally proves the similarity between the transfer function of the fixed part obtained theoretically and by identification.

Figure 10 shows the closed loop system consisting of the FO-PI controller given by relation (105) and theoretical transfer function given by relation (101), Bode diagram, Nyquist diagram and Nichols diagram. The stability of the system can be inferred from the specific interpretation of these diagrams and, in addition (to the conclusions in Figure 8), an amplitude stability margin of 12 dB can be noted.

For the implementation in DSP, it is necessary to obtain the transfer function of the FO-PI controller given in Equation (105) in an equivalent form as discrete variable $z$, but of integer order. For this, according to those presented in Section 2, an approximation of the fractional order transfer function can be obtained with an integer-order continuous transfer function, by using the Oustaloup filter.

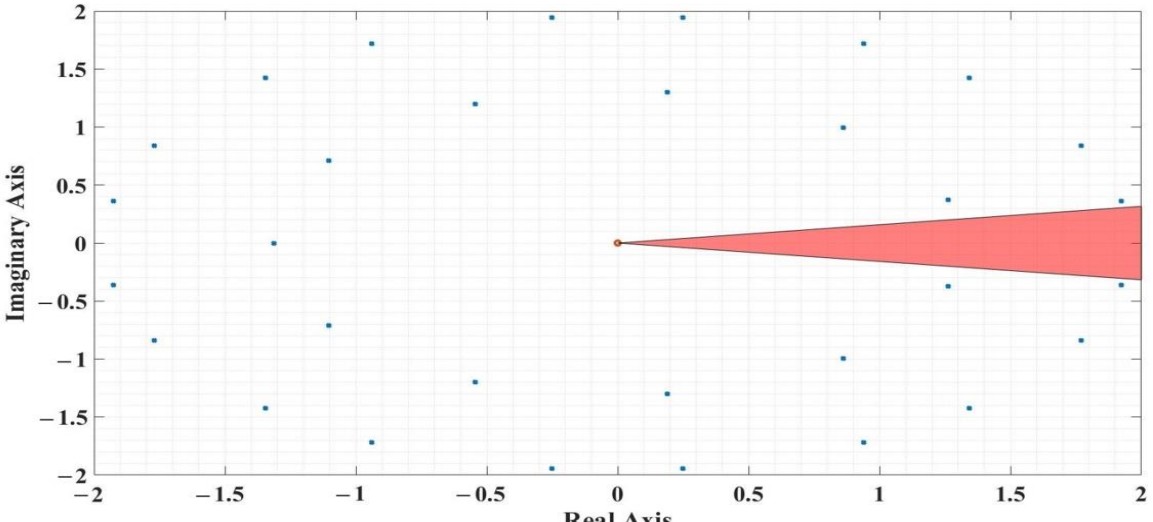

**Figure 8.** Graphical representation for the stability of the PMSM rotor speed control system—closed loop.

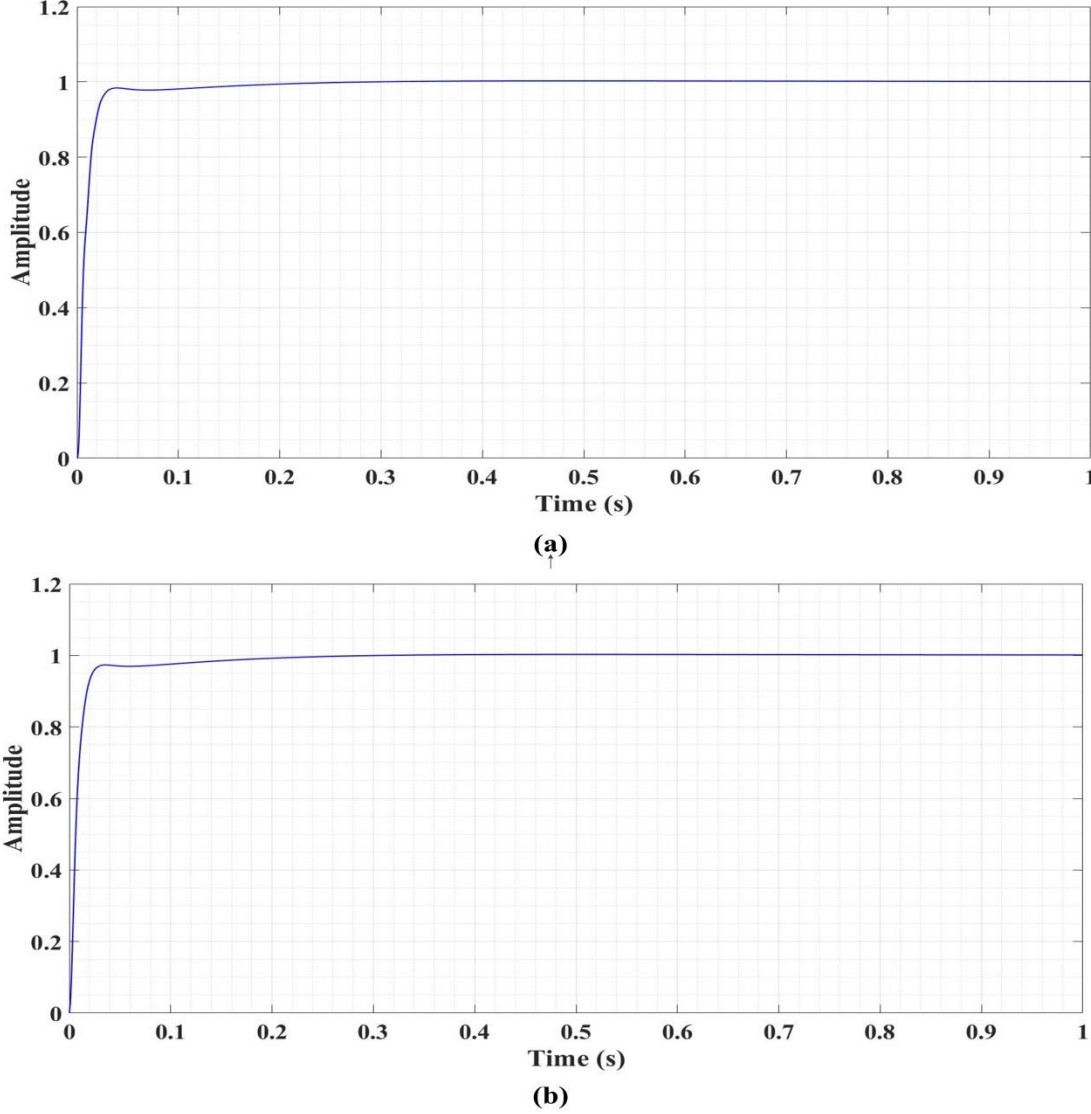

**Figure 9.** Unit step response of the PMSM rotor speed control system with FO-PI controller and fixed part with: (**a**) theoretical transfer function; (**b**) transfer function obtained through identification.

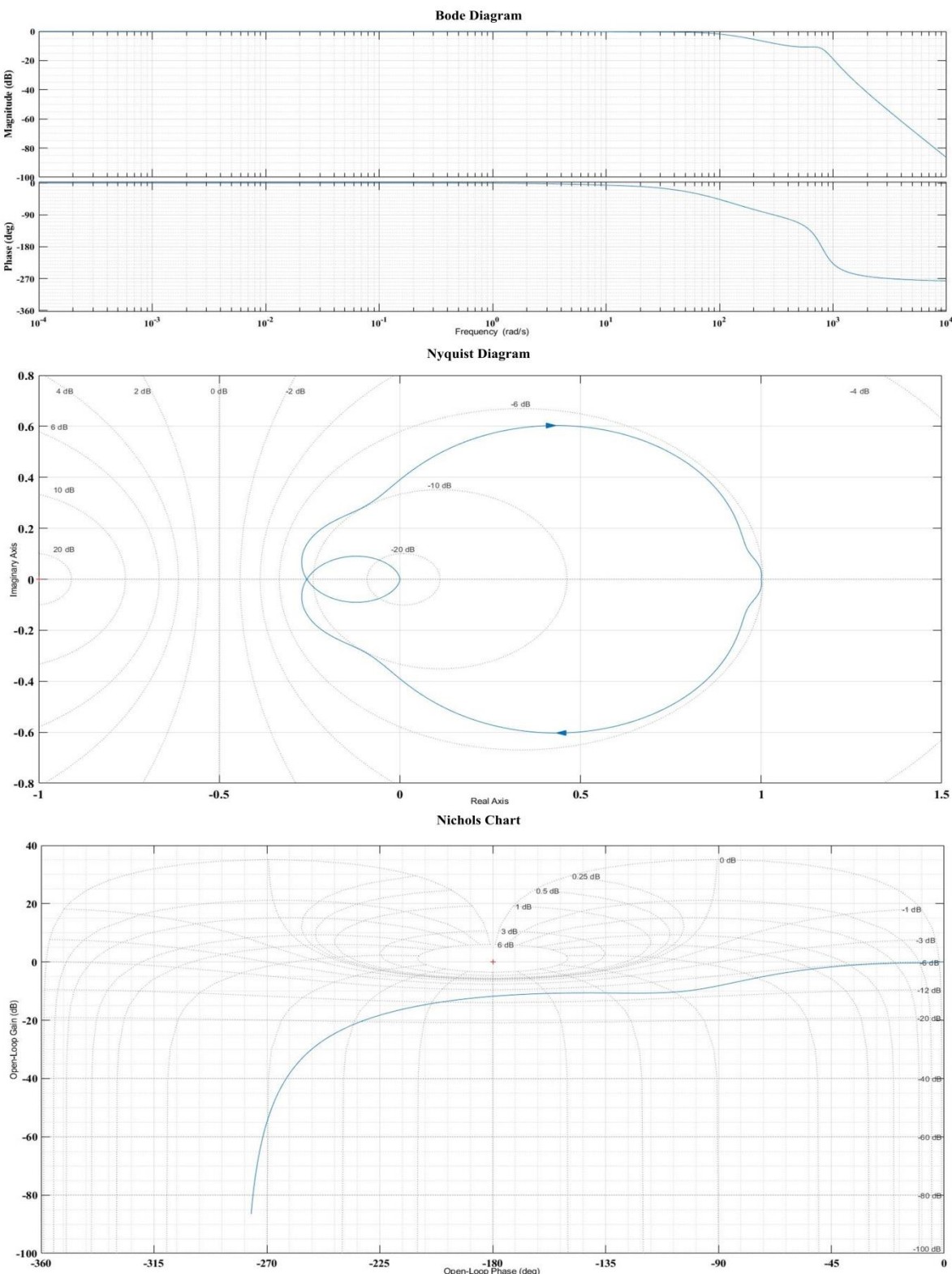

**Figure 10.** Bode, Nyquist and Nichols diagram representations for closed-loop system composed from FO-PI controller and theoretical transfer function of PMSM given by Equations (101) and (105).

In the usual frequency range for the presented application $\omega = (10^{-2}; 10^3)$ rad/s, in Equation (107) is expressed and the equivalent transfer function obtained. In order to obtain the discrete form of this

equivalent transfer function, the Tustin substitution is used, and the obtained discrete transfer function is expressed in Equation (108).

$$H_{PI\_INT}(s) = \frac{1.2(s+280.6)(s+354.8)(s+35.48)(s+25.88)(s+6.918)(s+4.422)}{s(s+354.8)(s+218.8)(s+35.48)(s+28.18)(s+3.548)} \cdot$$
$$\cdot \frac{(s+3.548)(s+0.3575)(s+0.3548)(s+0.0355)(s+0.03548)}{(s+0.3548)(s+0.2818)(s+0.03548)(s+0.02818)} \tag{107}$$

$$H_{PI\_INT}(z) = \frac{1.2031(z-0.7554)(z-0.7013)(z-0.9651)(z-0.9745)}{(z-1)^5(z-0.9972)(z-0.9965)(z-0.9722)} \cdot$$
$$\cdot \frac{1.2031(z-0.7554)(z-0.7013)(z-0.9651)(z-0.9745)}{(z-1)^5(z-0.9972)(z-0.9965)(z-0.9722)} \tag{108}$$

Another fractional order controller is presented in Equation (21) and for $K_t = 1.2$, $K_i = 12$, $n = 10$, and $K_d = 0$, the following transfer function of the fractional order TID controller is obtained:

$$H_{TID} = \frac{1.2 + 12s^{0.1}}{s^{1.1}} \tag{109}$$

The closed loop transfer function for the fixed part presented above in Equation (103) and the TID controller given by Equation (109) is the following:

$$H_{CL\_TID} = \frac{4.3272 \cdot 10^8 s^{0.91} + 4.3272 \cdot 10^9}{s^{3.1} + 7373s^{2.1} + 4.3746 \cdot 10^8 s^{0.91} + 4.3272 \cdot 10^9} \tag{110}$$

The response to the unit step of the PMSM rotor speed control loop with fractional TID controller is shown in Figure 11. A good performance is noted in dynamic and stationary regime.

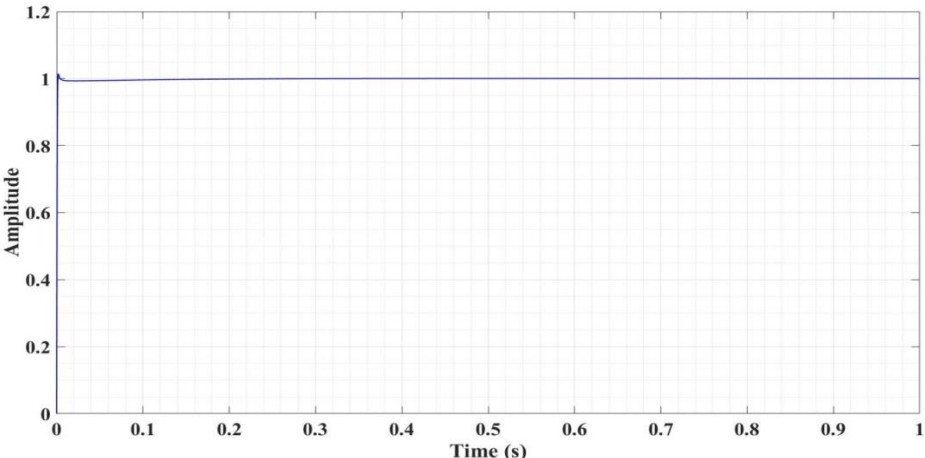

**Figure 11.** Unit step response of the PMSM rotor speed control system with transfer function obtained through identification for the fixed part and fractional TID controller.

Proceeding as above in the case of the FO-PI controller, using the Oustaloup filter zpk, in the TID controller case the results for integer-order will be as follows:

$$H_{TID\_INT}(s) = \frac{656.44(s+114.1)(s+17.33)(s+6.938)(s+0.8839)(s+0.08997)}{s(s+901.6)(s+90.16)(s+9.016)(s+0.9016)(s+0.09016)} \tag{111}$$

$$H_{TID\_INT}(z) = \frac{0.25176(z+0.7516)(z-0.8922)(z-9828)(z-0.9931)(z-1)}{(z-1)^3(z-0.991)(z-0.9138)(z-0.4059)} \tag{112}$$

In case of another fractional order controller which is presented in Equation (23), and for $k' = 300$, $x = 50$, $\lambda = 1.4$ and $\alpha = 0.11$, the following transfer function of the FO-lead-lag controller is obtained:

$$H_{LL} = \frac{3.606 \cdot 10^8}{s^{2.1} + 7.373 \cdot 10^4 s^{1.1} + 4.74 \cdot 10^6} \tag{113}$$

The closed loop transfer function for the fixed part presented above in Equation (103) and the FO-lead-lag controller given by Equation (113) is the following:

$$H_{CL\_LL} = \frac{abs\left(1.0561 \cdot 10^{11} - 6.942 \cdot 10^9 \cdot i\right)}{s^{2.1} + 73730 s^{1.1} + abs\left(1.0561 \cdot 10^{11} - 6.942 \cdot 10^9 \cdot i\right)} \tag{114}$$

The response to the unit step of the PMSM rotor speed control loop with the FO-lead-lag controller is shown in Figure 12. Furthermore, for this type of fractional order controller a good performance is noted in the dynamic and stationary regimes.

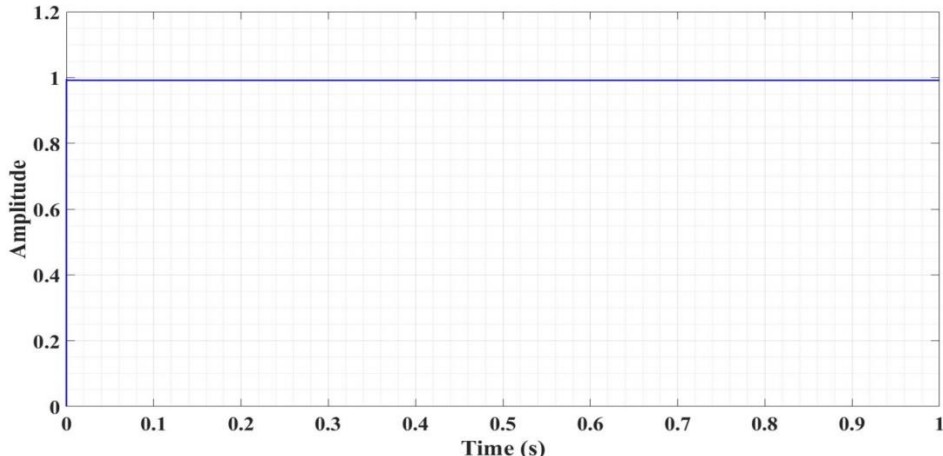

**Figure 12.** Unit step response of the PMSM rotor speed control system with transfer function obtained through identification for the fixed part and FO-lead-lag controller.

By using the Oustaloup filter zpk in the FO-lead-lag controller case the results for integer-order will be as follows:

$$H_{LL\_INT}(s) = \frac{1.8073 \cdot 10^8 (s + 354.8)^2 (s + 35.48)^2}{(s + 7.37 \cdot 10^4)(s + 354.8)(s + 272.4)(s + 35.48)(s + 3.603)(s + 3.548)} \cdot$$
$$\cdot \frac{(s + 3.548)^2 (s + 0.3548)^2 (s + 0.03548)^2}{(s + 0.3552)(s + 0.3548)(s + 0.03548)(s + 0.03548)(s^2 + 68.99 s + 1465)} \tag{115}$$

$$H_{LL\_INT}(z) = \frac{1.2103(z + 1.071)(z - 1)^4 (z - 0.9965)^2}{z(z - 1)^4 (z - 0.9965)(z - 0.9964)(z - 0.9651)} \cdot$$
$$\cdot \frac{(z - 0.9651)^2 (z - 0.7013)^2 (z + 0.0003655)}{(z - 0.7616)(z - 0.7013)(z^2 - 1.932 z + 0.9333)} \tag{116}$$

Figure 13 presents the results of the simulation of the PMSM rotor speed control, where the speed controller is FO-PI-type and PI-type, respectively. For a speed reference of 2000 rpm, with no load torque, good dynamic and static results are obtained, but with an obvious advantage of the FO-PI controller. The variations of the stator currents $i_{a,b,c}$, and of currents $i_d$ and $i_q$ are presented, and compliance with reference $i_{dref} = 0$ is noted according to the FOC strategy.

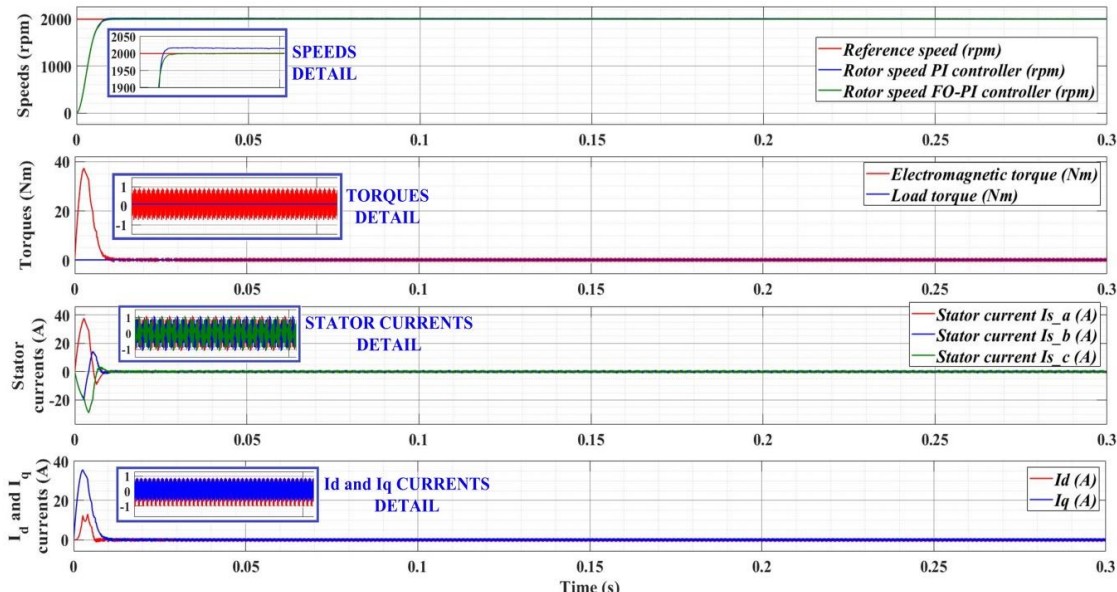

**Figure 13.** Comparative simulation of the PMSM time evolution with FO-PI speed controller and PI speed controller—$\omega_{ref}$ = 2000 rpm, $T_L$ = 0 Nm, $K_p$ = 1.2, $K_i$ = 12, $\lambda$ = 1.1 and $K_d$ = $\mu$ = 0.

In Figure 14, the FO-PI controller is compared to the PI controller for the PMSM rotor speed control system, for a speed reference of 2000 rpm and a load torque of 5 Nm. The overshooting by the speed PI-type controller and the good static and dynamic performance provided by the speed FO-PI-type controller can be noted in the detail in the said figure.

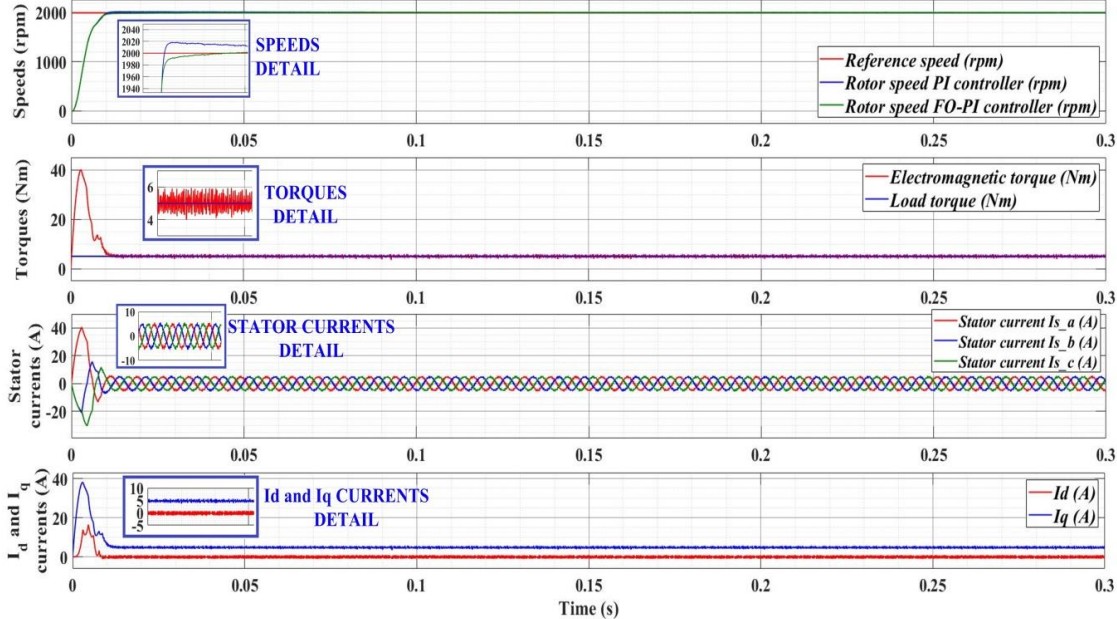

**Figure 14.** Comparative simulation of the PMSM time evolution with FO-PI speed controller and PI speed controller—$\omega_{ref}$ = 2000 rpm, $T_L$ = 5 Nm, $K_p$ = 1.2, $K_i$ = 12, $\lambda$ = 1.1 and $K_d$ = $\mu$ = 0.

The parametric robustness provided by the FO-PI controller for the PMSM rotor speed control loop is presented in Figure 15, where, for the speed and load torque references presented in Figure 14 plus a 50% increase of the *J* parameter (combined inertia of rotor and load), it is presented by maintaining a response with good dynamic and stationary performance.

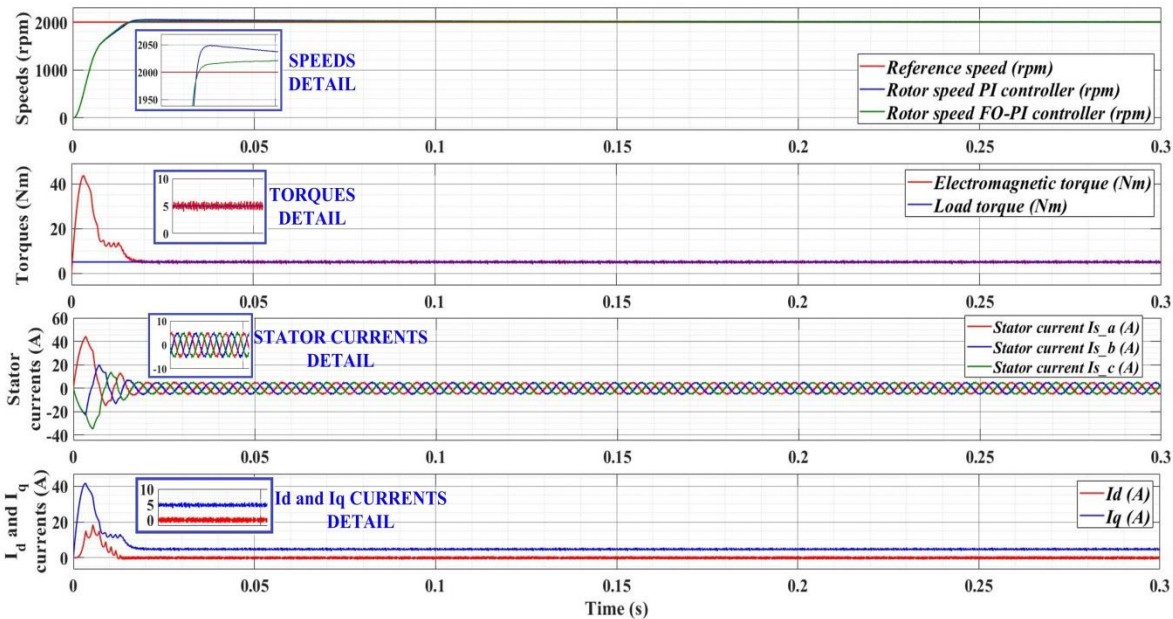

**Figure 15.** Comparative simulation of the PMSM time evolution with FO-PI speed controller and PI speed controller—$\omega_{ref}$ = 2000 rpm, $T_L$ = 5 Nm, $K_p$ = 1.2, $K_i$ = 12, $\lambda$ = 1.1, $K_d = \mu = 0$ and 100% increase of $J$ parameter.

Figure 16 shows the response of the control system in the case of $\omega_{ref}$ = 300 rpm and double the nominal load torque, $T_L$ = 10 Nm. Figure 16 also shows the time evolution of the electromagnetic torque, of the stator currents, as well as currents $i_d$ and $i_q$. A very good response time (12 ms) is noted, given the lack of overshooting and the evolution of the $i_d$ current around zero. For the implementation of the FO-SMC-type control described in Section 4, the parameters $\varepsilon$ = 300, $q$ = 200, $c$ = 100 and $\mu$ = 0.55 were selected.

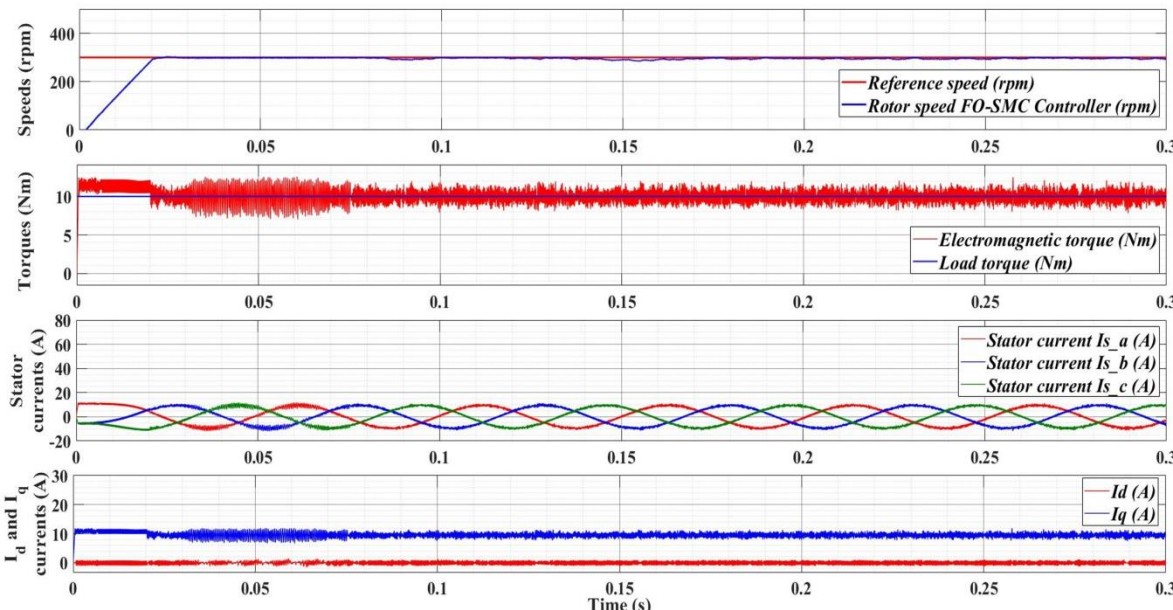

**Figure 16.** Simulation of the PMSM time evolution with FO-SMC speed controller for $\omega_{ref}$ = 300 rpm, $T_L$ = 10 Nm, $\varepsilon$ = 300, $q$ = 200, $c$ = 100 and $\mu$ = 0.55.

Figure 17 also shows the good performance of the PMSM control system using the FO-SMC-type controller for the outer rotor speed control loop if the speed reference $\omega_{ref}$ = 2200 rpm and the load torque $T_L$ = 2 Nm.

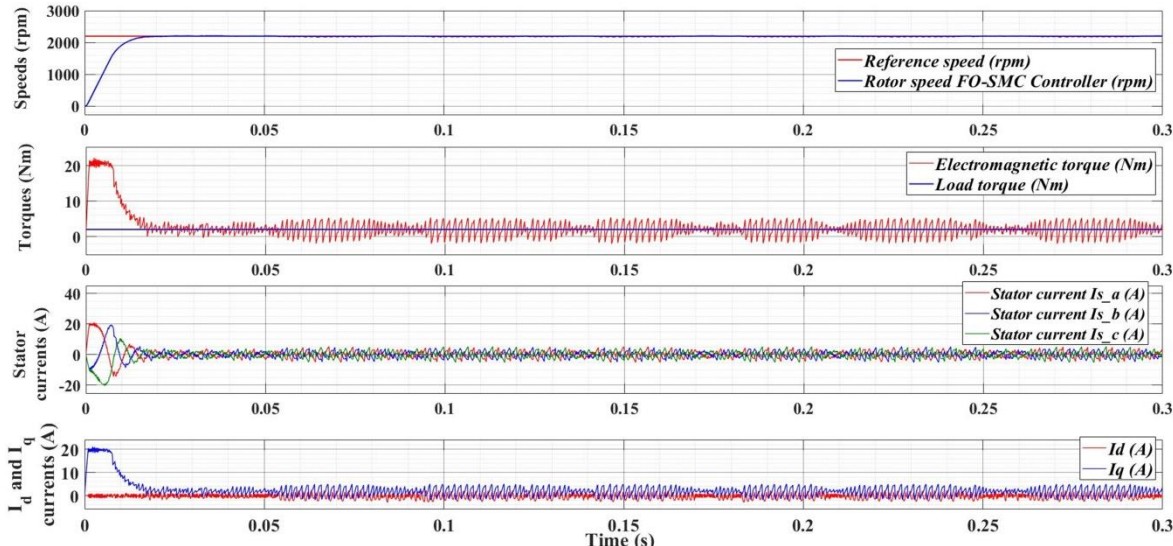

**Figure 17.** Simulation of the PMSM time evolution with FO-SMC speed controller for $\omega_{ref}$ = 2200 rpm, $T_L$ = 2 Nm, $\varepsilon$ = 300, $q$ = 200, $c$ = 100 and $\mu$ = 0.55.

The parametric robustness of the PMSM control system is noted in Figure 18, where, for the same speed and load torque references in Figure 17, however, with a 100% increase of $J$ parameter and an added uniformly distributed noise, good static and dynamic performances are noted, while there is an override of less than 2%.

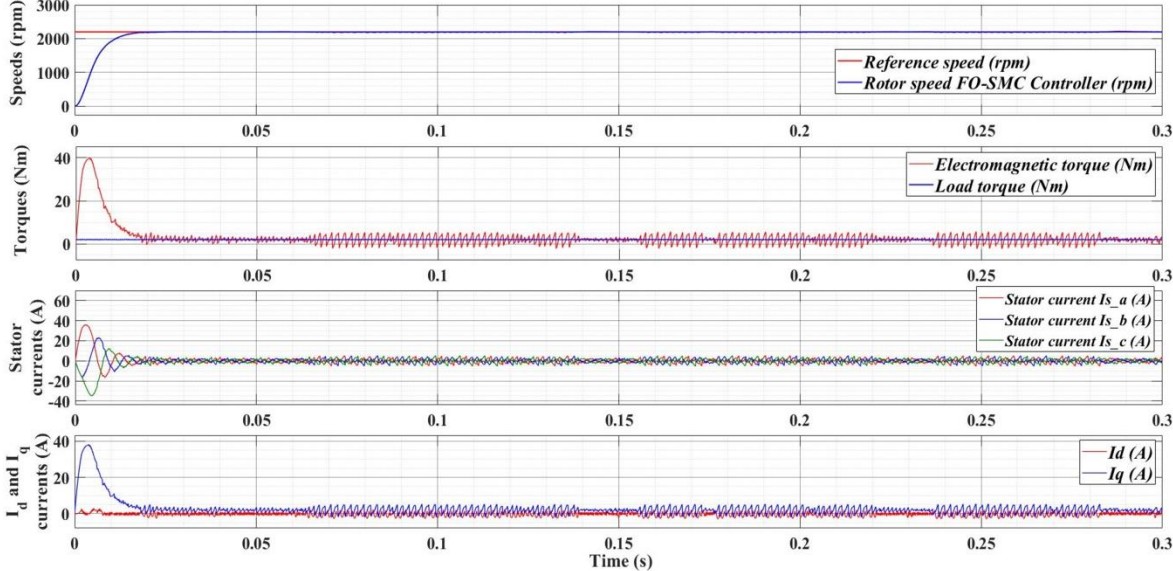

**Figure 18.** Simulation of the PMSM time evolution with FO-SMC speed controller for $\omega_{ref}$ = 2200 rpm, $T_L$ = 2 Nm and uniformly distributed noise, $\varepsilon$ = 300, $q$ = 200, $c$ = 100, $\mu$ = 0.55 and 100% increase of $J$ parameter.

For a comparison between the performance obtained for the PMSM rotor speed control by using the speed PI controller, the FO-PI speed controller, the TID speed controller, the FO-lead-lag speed controller and the FO-SMC speed controller, Figure 19 shows the system response in closed loop for a reference of 300 rpm and a load torque of 10 Nm. The superiority of the speed FO-PI-type speed controller, TID-type speed controller, FO-lead-lag speed controller and FO-SMC speed controller over the speed PI-type controller is noted. This can be intuited by the fact that the first two controllers mentioned have a higher number of tuning parameters than the PI-type controller, and their mathematical

model can be considered more accurate due to the use of the fractional calculus for integration and differentiation operators. Of the four speed controllers compared, the superiority is apparent for the speed FO-SMC-type controller which does not feed any overshooting, and the response time is below 15 ms for the nominal load torque.

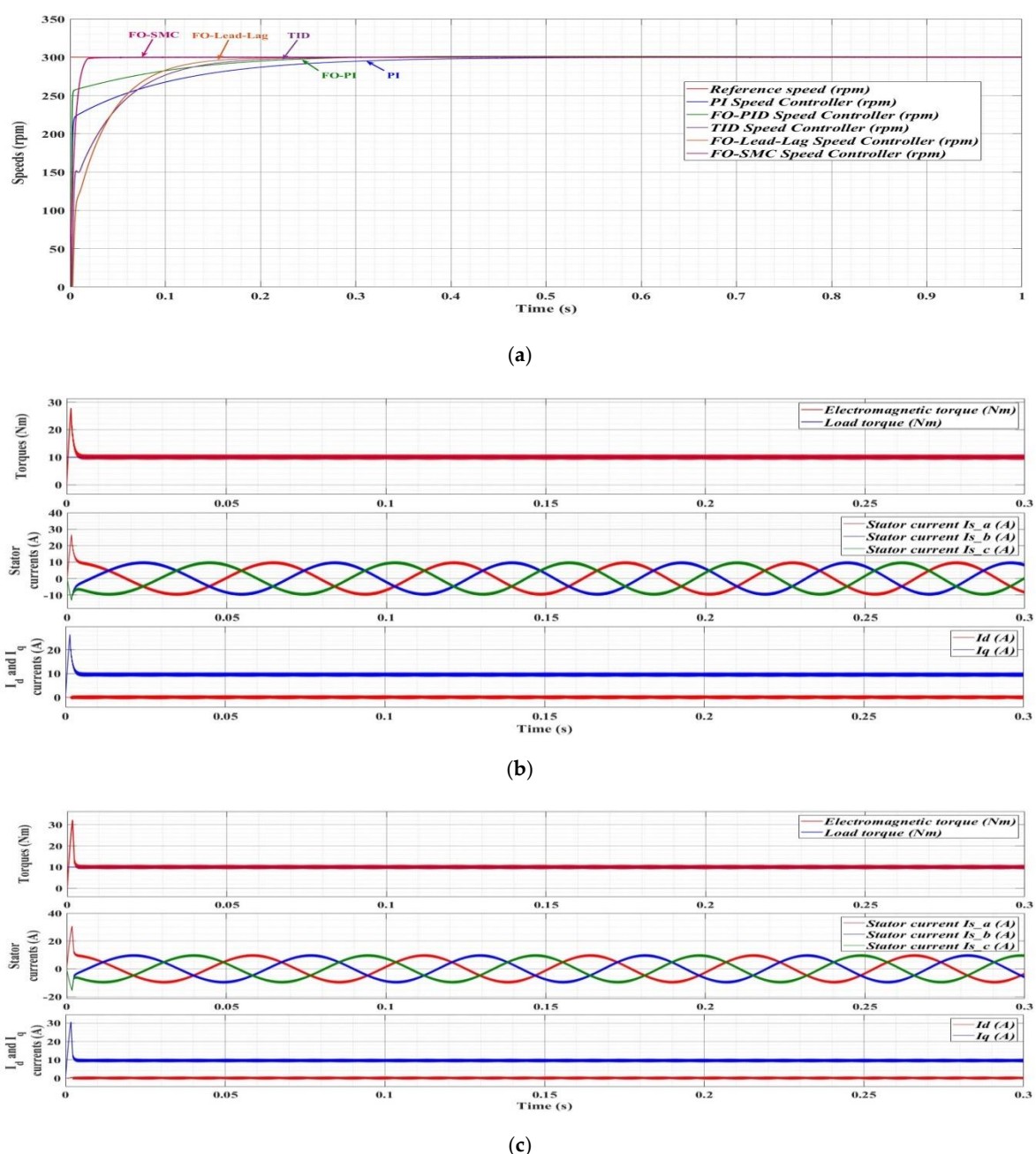

**Figure 19.** *Cont.*

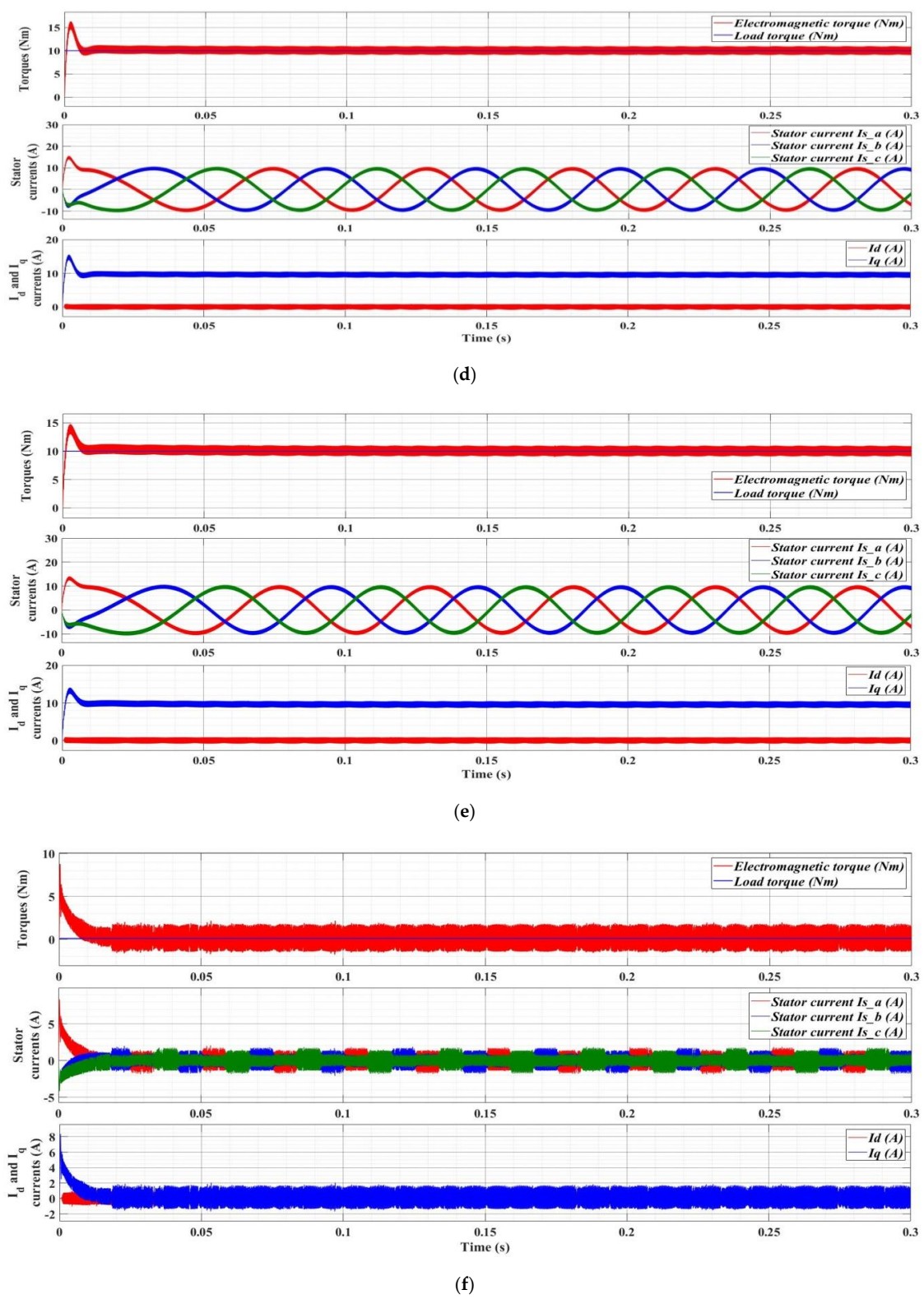

**Figure 19.** Comparative simulation of the PMSM time evolution with FO-SMC speed controller, FO-lead-lag speed controller, TID speed controller, FO-PI speed controller and PI speed controller—$\omega_{ref}$ = 300 rpm, $T_L$ = 10 Nm: (**a**) speed comparison; (**b**) torques, stator currents and $i_d$ and $i_q$ currents for PI speed controller; (**c**) torques, stator currents and $i_d$ and $i_q$ currents for FO-PI speed controller; (**d**) torques, stator currents and $i_d$ and $i_q$ currents for TID speed controller; (**e**) torques, stator currents and $i_d$ and $i_q$ currents for FO-lead-lag speed controller; and (**f**) torques, stator currents and $i_d$ and $i_q$ currents for FO-SMC speed controller.

The value of the speed/torque ripple is defined as follows:

$$x_{rip} = \sqrt{\frac{1}{N} \sum_{i=1}^{N} \left( x(i) - x_{ref}(i) \right)^2} \qquad (117)$$

where $N$ represents the number of samples, $x$ and $x_{ref}$ represent the rotor speed/torque and the prescribed reference of speed/torque, respectively.

Table 2 compares a number of performance indices obtained by using fractional speed controllers for a reference speed $\omega_{ref}$ = 300 rpm and a load torque reference $T_L$ = 10 Nm. The performances of these types of controllers are presented both in the case of nominal parameters, and in the case of *J* parameter doubling. It can be noted that the fractional controllers and especially the FO-SMC speed controller have superior performance.

**Table 2.** Comparison of performance indices of the fractional order proposed speed controllers.

| Performance Indices | PI Speed Controller | FO-PI Speed Controller | TID Speed Controller | FO-Lead-Lag Speed Controller | FO-SMC Speed Controller |
|---|---|---|---|---|---|
| Overshoot (%) nominal *J* | 0 | 0 | 0 | 0 | 0 |
| Overshoot (%) double *J* | 0 | 0 | 0 | 0 | 0 |
| Settling time (ms) nominal *J* | 300 | 240 | 220 | 180 | 16 |
| Settling time (ms) double *J* | 450 | 300 | 280 | 220 | 40 |
| Steady state error (%) nominal *J* | 0.11 | 0.11 | 0.1 | 0.1 | 0.09 |
| Steady state error (%) double *J* | 0.12 | 0.12 | 0.11 | 0.1 | 0.09 |
| Speed ripple (rpm) nominal *J* | 121.78 | 81.14 | 142.24 | 112.94 | 102.81 |
| Speed ripple (rpm) double *J* | 289.28 | 192.35 | 216.14 | 204.91 | 131.15 |
| Torque ripple (Nm) nominal *J* | 13.68 | 17.16 | 10.08 | 9.23 | 18.91 |
| Torque ripple (Nm) double *J* | 11.8 | 15.45 | 12.91 | 12.1 | 16.01 |

*7.2. Numerical Simulations for Rotor Speed Estimation and Fault Detection*

Figure 20 shows the evolution of the SMO-type observer for the estimation of the PMSM rotor speed. Very good stationary results are noted, except for the first 100 ms, when, normally, at the start of PMSM, it is controlled in the open loop, according to a predefined sequence.

The numerical simulation parameters $L_1$ = 150,000 and $L_2$ = 50 (amplification factors of the FDO-type observer) are selected to test the efficiency of the FDO-type observer described in Section 6. Thus, Figure 21 shows the efficiency of this type of observer due to the fact that it very precisely reconstructed the currents $i_\alpha$ and $i_\beta$ together with the outputs $z_\alpha$ and $z_\beta$ provided by Equation (84), which describes the implementation of the FDO-type observer under the conditions where faults of the current sensors may occur.

Based on Equations (91)–(93), which express the relationships between the fault flags on phases $\alpha$ and $\beta$ in the $\alpha$-$\beta$ reference frame and the real supply phases a, b, c of the PMSM, Figure 22 shows that, for the occurrence of a fault on phase a, it is detected, and a specific fault flag is set to logic 1, during the occurrence of such a fault of the current sensor. The detection threshold used is 4 A, and a fast response of the FDO-type observer of 60 ms is noted.

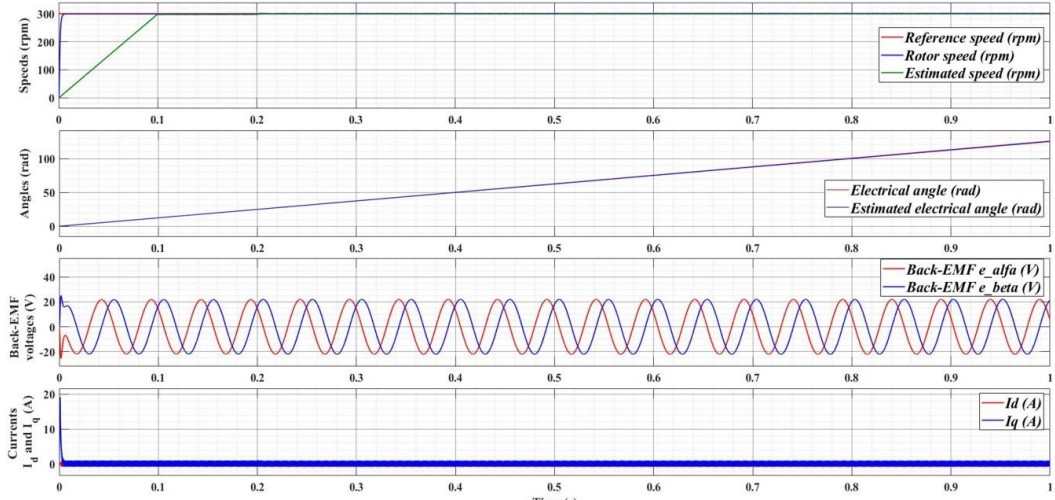

**Figure 20.** Simulation of the back-EMF, rotor position and rotor speed time evolution from the SMO-type observer.

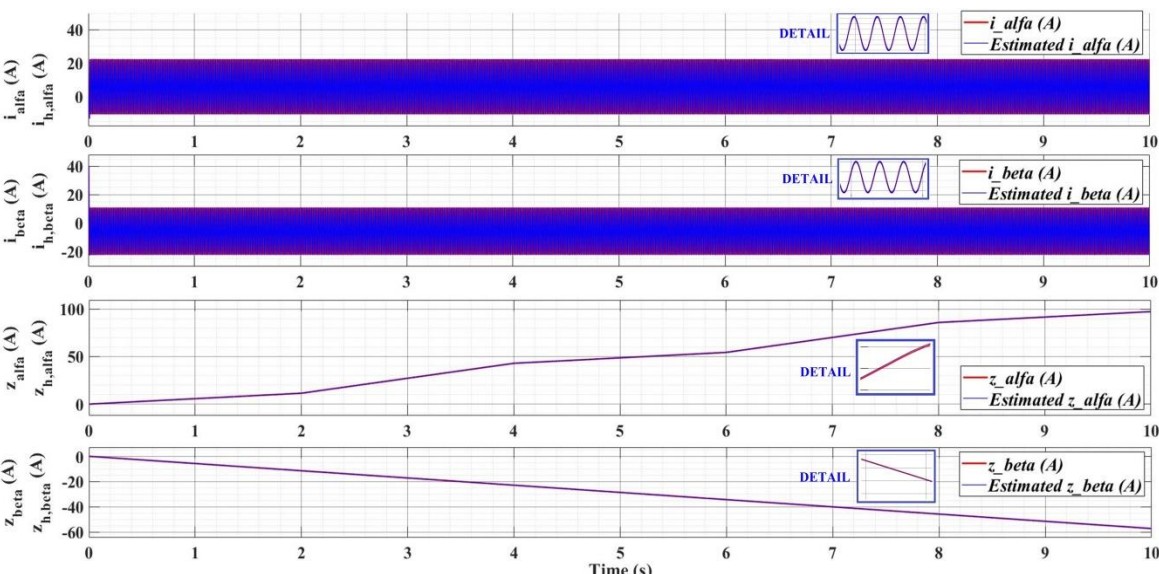

**Figure 21.** Estimated signals time evolution based on FDO-type observer.

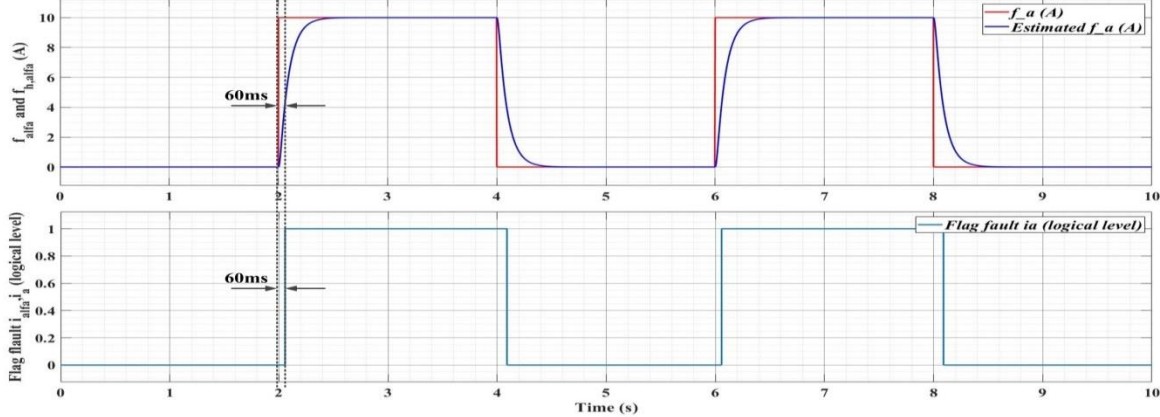

**Figure 22.** Estimated fault on phase "a" based on FDO-type observer.

The response time of the FDO-type observer is the time between the moment of occurrence of a fault of the sensor and the setting of the fault flag, under the conditions where the occurrence of false fault detections is eliminated by setting of the 4 A threshold, and the disturbance on the system ($d_1$ and $d_2$, described in Equation (80) used the in this numerical simulation is a uniform random variable with amplitude in the range −10 to 10 A.

### 7.3. Numerical Simulations—Fractional Order Synergetic Current Controllers and PI Speed Controller for PMSM

Figure 23 presents the block diagram of the MATLAB/Simulink implementation of the PMSM sensorless control system based on the synergetic current control or the FO-synergetic current control for the inner current loops control.

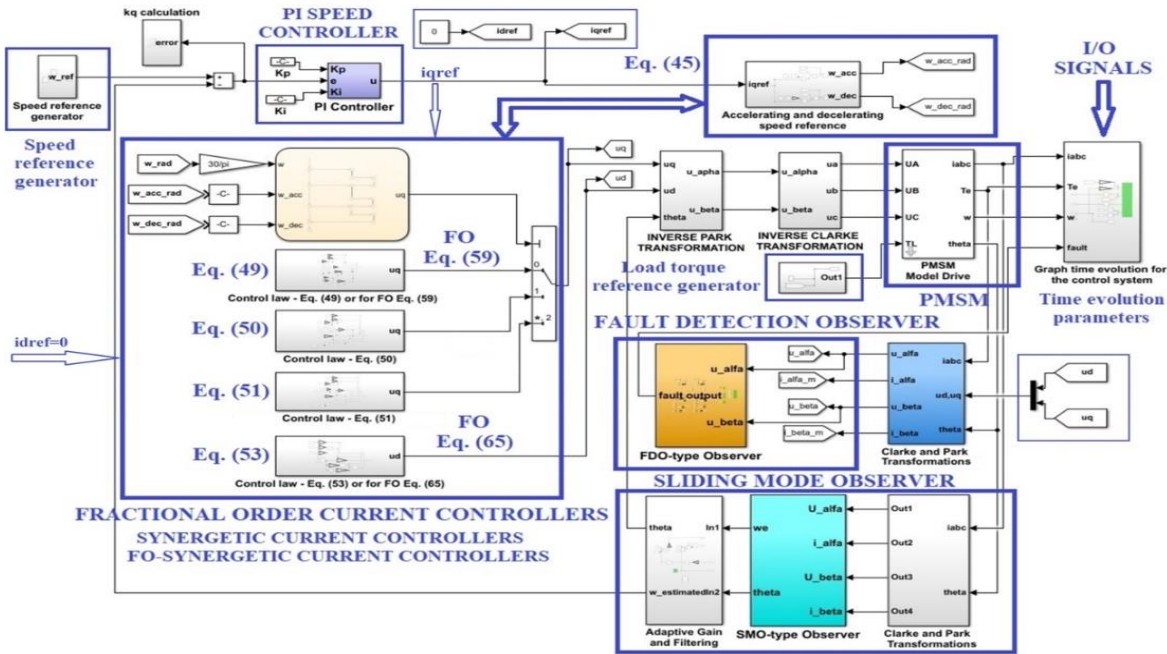

**Figure 23.** Simulink block diagram for the PMSM sensorless control based on PI speed controller, FO-synergetic current controllers, SMO speed observer and FDO.

The main functional blocks represented in the block diagram of the control system are: the PI speed controller which supplies current $i_{qref}$ ($i_{dref} = 0$ according to the FOC control strategy), the speed reference generator, the synergetic controller or the FO-synergetic controller, the load torque generator, the PMSM motor drive, the SMO-type observer, the acceleration and deceleration of the rotor angular velocity and the Clarke and Park transformation.

The synergetic current controllers or the FO-synergetic current controller block provides the controls $u_d$ and $u_q$ by implementing Equations (49)–(51) and (53) for the synergetic current control, and Equations (59) and (65) for the FO-synergetic current control, respectively.

For a speed reference of 500 rpm and a torque load of 1 Nm when using a synergetic controller, Figure 24 presents the qualitatively superior response of the system with an override below 8%, but with a notable performance response time of 2 ms.

For the numerical simulations in which the FO-synergetic type controller is used, the parameters described in Section 5 are: $k_{iq} = 10,000$, $k_q = 10,000$, $i_{qmax} = 50$, $T_d = 3$, $T_q = 3$, $k_{id} = 10,000$ and $\mu = 0.5$ (for Equations (54) and (60)).

In Figure 25, by replacing the synergetic controller with the FO-synergetic controller, for the same conditions presented in Figure 24, the reduction of the override to 4% and an excellent performance determined by the 1 ms response time are noted.

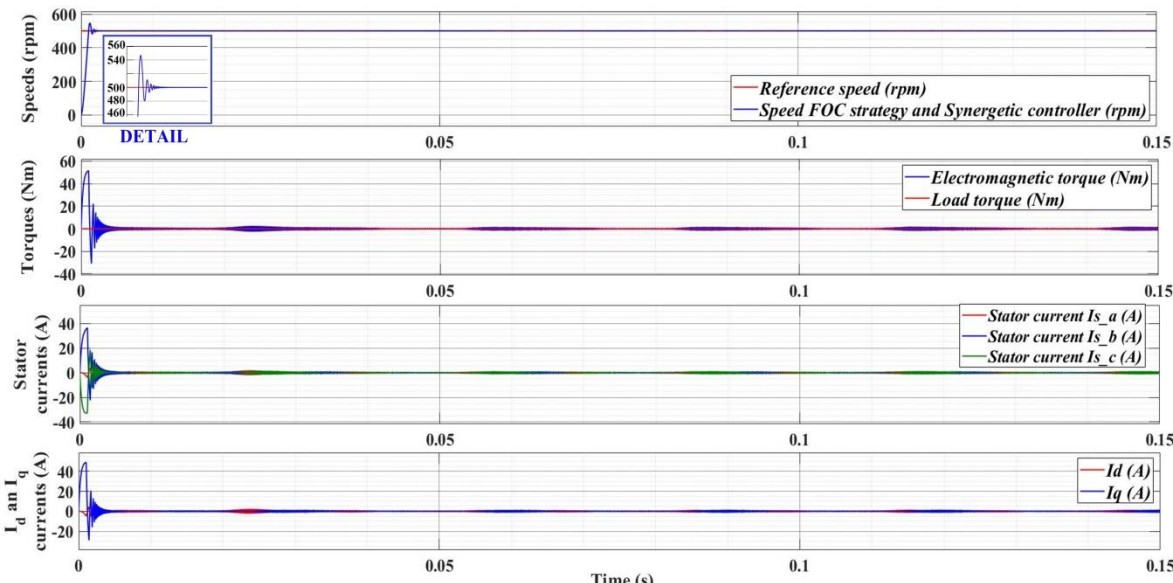

**Figure 24.** Simulation of the PMSM time evolution with synergetic controller and FOC strategy— $\omega_{ref}$ = 500 rpm, $T_L$ = 1 Nm, $k_{iq}$ = 10,000, $k_q$ = 10,000, $i_{qmax}$ = 50, $T_d$ = 3, $T_q$ = 3 and $k_{id}$ = 10,000.

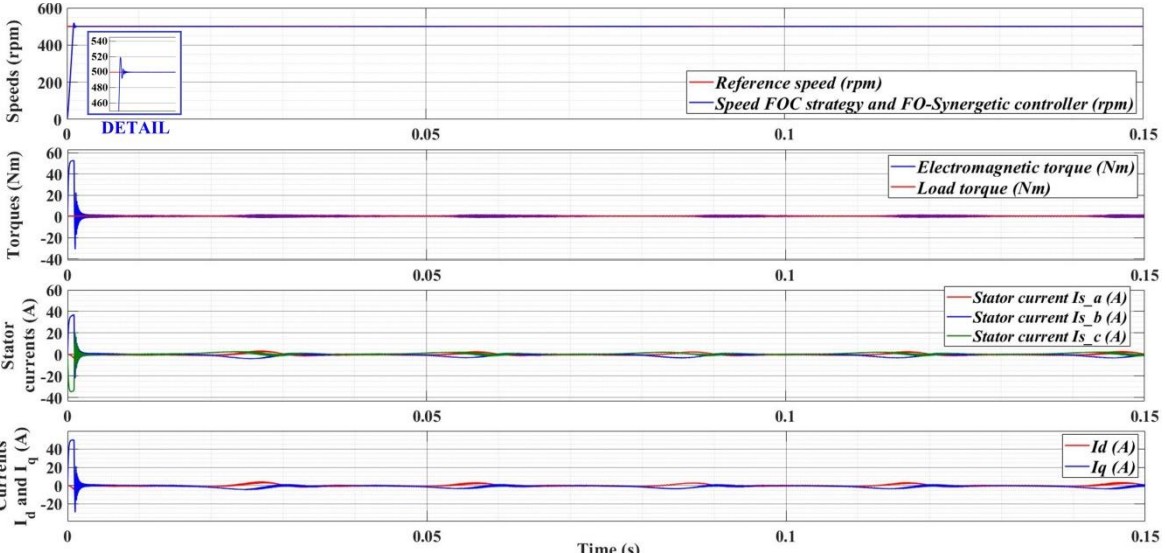

**Figure 25.** Simulation of the PMSM time evolution with FO-synergetic controller, and FOC strategy— $\omega_{ref}$ = 500 rpm, $T_L$ = 1 Nm, $k_{iq}$ = 10,000, $k_q$ = 10,000, $i_{qmax}$ = 50, $T_d$ = 3, $T_q$ = 3, $k_{id}$ = 10,000 and $\mu$ = 0.5.

Figures 24 and 25 additionally present the evolution of the electromagnetic torques, load torques and stator currents $i_a$, $i_b$, $i_c$ and currents $i_d$ and $i_q$.

Figure 26 presents the comparative time evolution of the numerical simulation for the FOC strategy with PI current controllers, synergetic current controllers and FO-synergetic current controllers of the PMSM. Figure 26 shows that very good performance achieved by using the synergetic control for the inner current loops $i_d$ and $i_q$. Obviously, due to the additional control parameters, between the two types of synergetic control, the best performance is obtained by using the FO-synergetic current controllers.

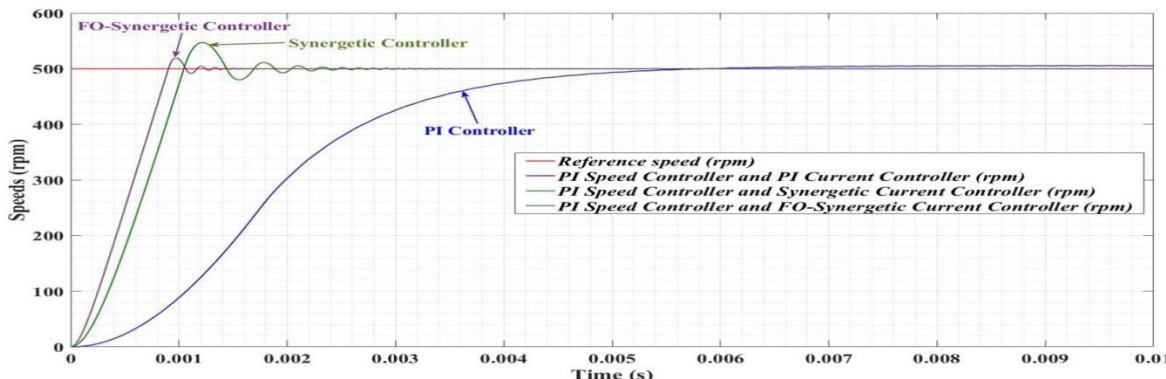

**Figure 26.** Comparative time evolution of the numerical simulation for FOC strategy with PI controller, synergetic controller and FO-synergetic controller of the PMSM—$\omega ref$ = 500 rpm and $T_L$ = 1 Nm.

Table 3 compares a number of performance indices obtained by using fractional current controllers for a reference speed $\omega_{ref}$ = 500 rpm and a load torque reference $T_L$ = 1 Nm. The performances of these types of controllers are presented both in the case of nominal parameters, and in the case of $J$ parameter doubling. It can be noted that the synergetic controllers, and especially the FO-synergetic current controller, have superior performance.

**Table 3.** Comparison of performance indices of the fractional order proposed current controller.

| Performance Indices | PI Current Controller | Synergetic Current Controller | FO-Synergetic Current Controller |
|---|---|---|---|
| Overshoot (%) nominal $J$ | 0 | 8 | 2 |
| Overshoot (%) double $J$ | 0 | 14 | 3.5 |
| Settling time (ms) nominal $J$ | 6 | 1.2 | 1 |
| Settling time (ms) double $J$ | 11 | 1.8 | 1.6 |
| Steady state error (%) nominal $J$ | 0.1 | 0.08 | 0.07 |
| Steady state error (%) double $J$ | 0.1 | 0.08 | 0.07 |
| Speed ripple (rpm) nominal $J$ | 182.16 | 112.91 | 102.45 |
| Speed ripple (rpm) double $J$ | 214.91 | 129.54 | 107.63 |
| Torque ripple (Nm) nominal $J$ | 15.21 | 17.95 | 15.82 |
| Torque ripple (Nm) double $J$ | 19.44 | 21.02 | 18.73 |

*7.4. Numerical Simulations—Fractional Order Speed Controllers and Fractional Order Synergetic Current Controller for PMSM*

This subsection presents the numerical simulations for the structure proposed in this article, namely, an FOC control strategy of the PMSM where the speed controller is of the SMC or FO-SMC types, and the controllers for the current control loops are of synergetic and FO-synergetic type. Figure 27 shows the block diagram of the MATLAB/Simulink implementation.



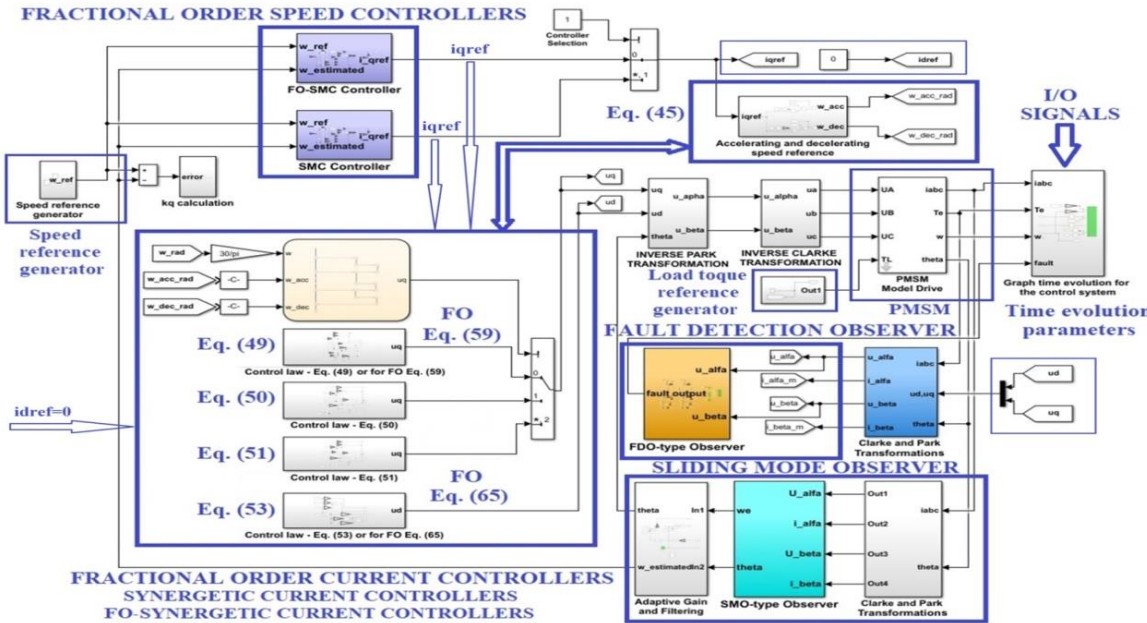

**Figure 27.** Simulink block diagram for the PMSM sensorless control based FO-SMC speed controller, FO-synergetic current controller, SMO speed observer and FDO.

For the implementation of the FO-SMC-type control described in Section 4, the parameters $\varepsilon = 300$, $q = 200$, $c = 100$ and $\mu = 0.55$ were selected, and for the FO-synergetic type controller described in Section 5, the parameters $k_{iq} = 10{,}000$, $k_q = 10{,}000$, $i_{qmax} = 50$, $T_d = 3$, $T_q = 3$, $k_{id} = 10{,}000$ and $\mu = 0.5$, for Equations (54) and (60), were selected.

The evolution of the time response for the PMSM control system with FO-SMC speed controller and FO-synergetic current controller for $\omega_{ref} = 500$ rpm, $T_L = 1$ Nm is presented in Figure 28. Very good static and dynamic performances can be noted, of which we mention the response time of 0.92 ms and an overshoot of less than 1.2%.

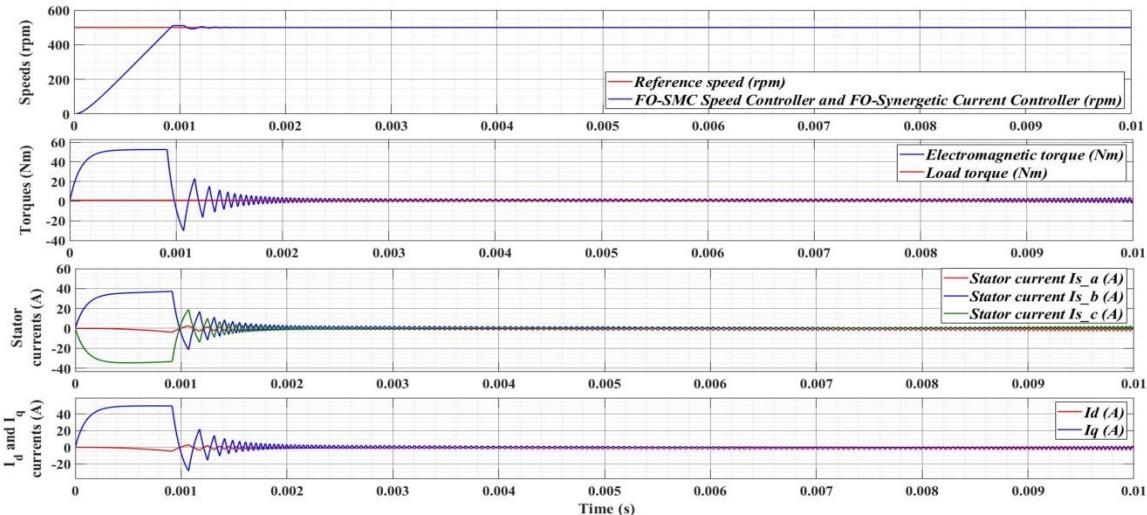

**Figure 28.** Simulation of the PMSM time evolution with FO-SMC speed controller and FO-synergetic current controller—$\omega_{ref} = 500$ rpm, $T_L = 1$ Nm, ($\varepsilon = 300$, $q = 200$, $c = 100$ and $\mu = 0.55$ for FO-SMC speed controller), ($k_{iq} = 10{,}000$, $k_q = 10{,}000$, $i_{qmax} = 50$, $T_d = 3$, $T_q = 3$, $k_{id} = 10{,}000$ and $\mu = 0.5$ for FO-synergetic current controllers).

In Figure 29, under the same conditions, but for a load torque of 10 Nm, very good performances of the PMSM control system with a response time of 1.2 ms are also obtained. The parametric robustness

of the proposed control system is demonstrated in Figure 30 by obtaining a response time of 1.6 ms under the conditions where a uniformly distributed noise with 0.2 Nm magnitude additionally acts on the load torque, $J$ parameter has a 100% increase, and also the stator resistance $R_s$ has double the nominal value.

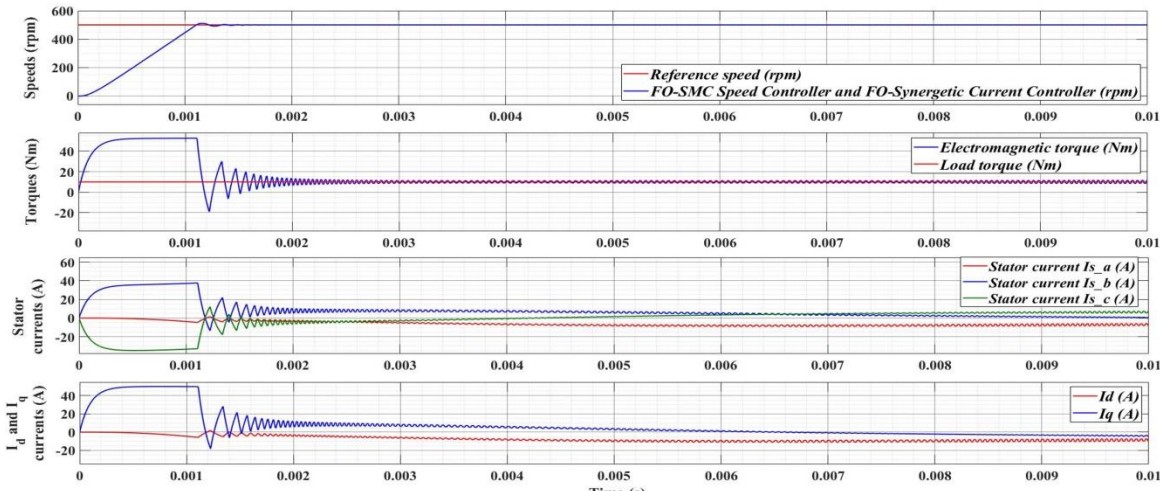

**Figure 29.** Simulation of the PMSM time evolution with FO-SMC speed controller and FO-synergetic current controller—$\omega_{ref}$ = 500 rpm, $T_L$ = 10 Nm, ($\varepsilon$ = 300, $q$ = 200, $c$ = 100 and $\mu$ = 0.55 for FO-SMC speed controller), ($k_{iq}$ = 10,000, $k_q$ = 10,000, $i_{qmax}$ = 50, $T_d$ = 3, $T_q$ = 3, $k_{id}$ = 10,000 and $\mu$ = 0.5 for FO-synergetic current controllers).

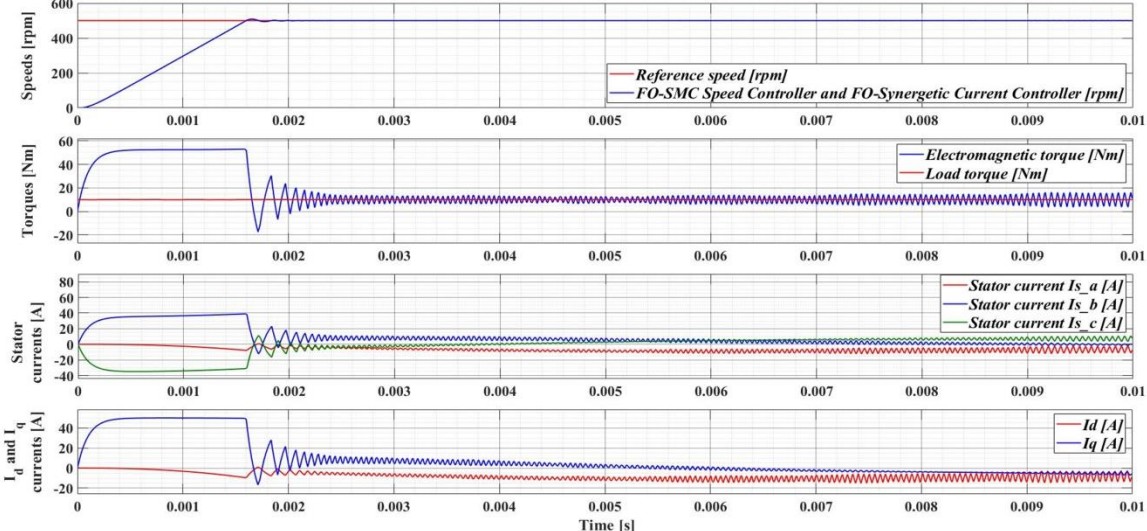

**Figure 30.** Simulation of the PMSM time evolution with FO-SMC speed controller and FO-synergetic current controller—$\omega_{ref}$ = 500 rpm, $T_L$ = 10 Nm and uniformly distributed noise, ($\varepsilon$ = 300, $q$ = 200, $c$ = 100 and $\mu$ = 0.55 for FO-SMC speed controller), ($k_{iq}$ = 10,000, $k_q$ = 10,000, $i_{qmax}$ = 50, $T_d$ = 3, $T_q$ = 3, $k_{id}$ = 10,000 and $\mu$ = 0.5 for FO-synergetic current controllers), 100% increase of $J$ parameter and 100% increase of stator resistance $R_s$.

Figure 31 compares the response of four PMSM control systems obtained by combinations of SMC and FO-SMC speed control systems, and the current controllers are of synergetic and FO-synergetic type. Table 4 presents the comparative results of the performance of these control systems according to: overshoot, settling time, steady state error and speed ripple defined in relation (117). It is obvious that the control system proposed in this article based on FO-SMC speed controller and FO-synergetic current controllers has the best performance.

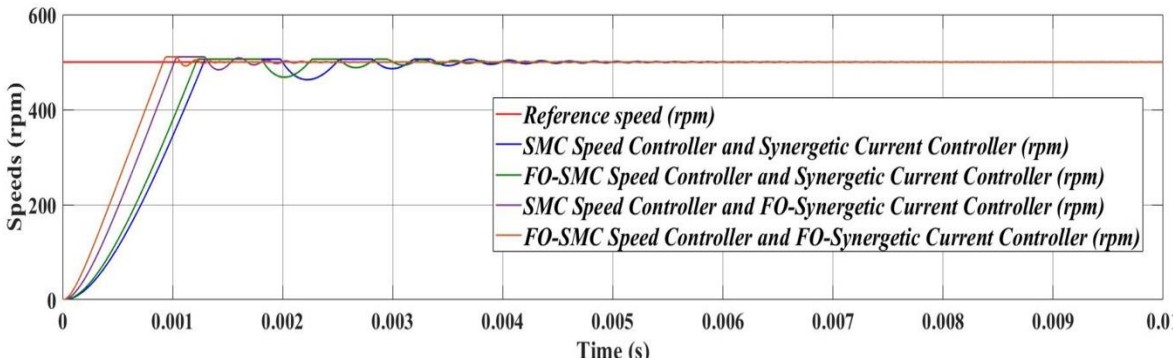

**Figure 31.** Simulation of the PMSM time evolution comparative fractional order controllers—$\omega_{ref}$ = 500 rpm and $T_L$ = 1 Nm.

**Table 4.** Comparison of performance indices of the fractional order proposed controllers.

| Performance Indices | SMC Speed Controller and Synergetic Currents Controller | FO-SMC Speed Controller and Synergetic Currents Controller | SMC Speed Controller and FO-Synergetic Currents Controller | FO-SMC Speed Controller and FO-Synergetic Currents Controller |
|---|---|---|---|---|
| Overshoot (%) nominal $J$ | 1.15 | 1.15 | 1.18 | 1.15 |
| Overshoot (%) double $J$ | 1.18 | 1.9 | 1.8 | 1.2 |
| Settling time (ms) nominal $J$ | 1.4 | 1.3 | 1 | 0.92 |
| Settling time (ms) double $J$ | 1.5 | 1.7 | 1.22 | 1.8 |
| Steady state error (%) nominal $J$ | 0.07 | 0.07 | 0.06 | 0.06 |
| Steady state error (%) double $J$ | 0.08 | 0.07 | 0.06 | 0.06 |
| Speed ripple (rpm) nominal $J$ | 123.03 | 118.73 | 104.50 | 95.34 |
| Speed ripple (rpm) double $J$ | 149.25 | 148.16 | 120.85 | 83.09 |
| Torque ripple (Nm) nominal $J$ | 14.92 | 14.74 | 126.29 | 14.71 |
| Torque ripple (Nm) double $J$ | 20.96 | 20.82 | 112.13 | 14.33 |

It is obviously noticeable that the values obtained for the settling time are 0.92 ms under nominal parameters of the PMSM, along with the other performance indices which can be considered as very good for the FO-SMC speed controller and the FO-synergetic current controller. The PMSM used in these numerical simulations is implemented in Power Systems/Simscape Electrical toolbox from Simulink, and in many scientific papers it is used as benchmark, and, to our best knowledge, the settling time of 0.92 ms obtained when using the FO-SMC speed controller and the FO-synergetic current controller is the best settling time obtained for a usual range of the speed reference and load torque, and for any other proposed controllers.

In addition, under the same conditions as a classic FOC-type control system of a PMSM, Figure 32 also shows an improvement in the THD of the currents in the PMSM supply phases. Thus, THD is reduced from 50% to 22%. Obviously, for further reduction of the THD, additional specialized systems as those presented in [42] can also be added.

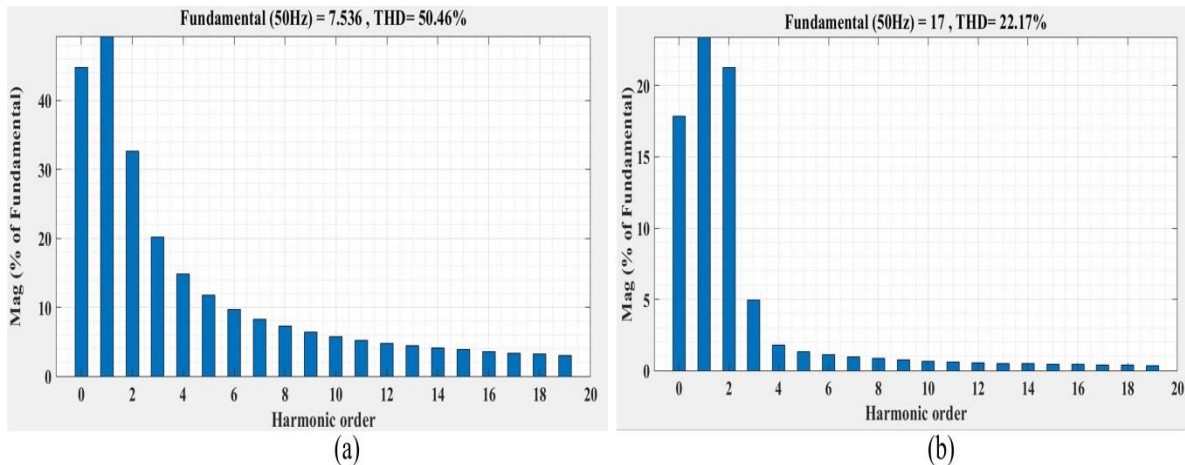

**Figure 32.** THD current analysis: (**a**) PMSM control system with PI speed controller and PI current controllers; (**b**) PMSM control system with FO-SMC speed controller and FO-synergetic current controllers.

## 8. Experimental Results

For the experimental validation of the simulations of the proposed PMSM control algorithms, which were presented and validated by numerical simulations in the previous section, this section presents the development platform used for their real time implementation in embedded systems.

The developed algorithms are implemented in MATLAB/Simulink and use dedicated functions from specialized libraries in the Model-Based Design Toolbox S32K1xx Series which contains the Automotive Math and Motor Control Library Set for NXP S32K14x devices, a toolbox which is dedicated to the PMSM control.

The hardware platform dedicated for the PMSM control for the experimental testing of the proposed algorithms is an S32K144 development kit which contains an S32K144 evaluation board (S32K144EVB-Q100), DEVKIT-MOTORGD board based on SMARTMOS GD3000 pre-driver and Linix 45ZWN24-40 PMSM type. The controller of the development platform is S32K144 MCU which is of 32bit Cortex M4F type, which has a time base of 112 MHz with 512 KB of flash memory and 54 KB of RAM.

There are also a number of dedicated hardware peripherals (FTM, ADC, PDB, PWM, timers) for the PMSM control, common analog and digital I/O processing blocks, but also a wide range of communication blocks which are common for the industrial environment. Among the communication interfaces we mention OpenSDA serial debug interface, CAN controller with CAN-FD protocol.

Figure 33 shows an image of the experimental platform.

The software application block diagram of the implementation in MATLAB/Simulink and Model-Based Design Toolbox S32K1xx Series NXP for the embedded system of the PMSM control system is presented in Figure 34. The main blocks presented are: data acquisition and commands, which is supervised by the dispatcher software interrupters, current controllers from inner loop and speed controller from outer loop.

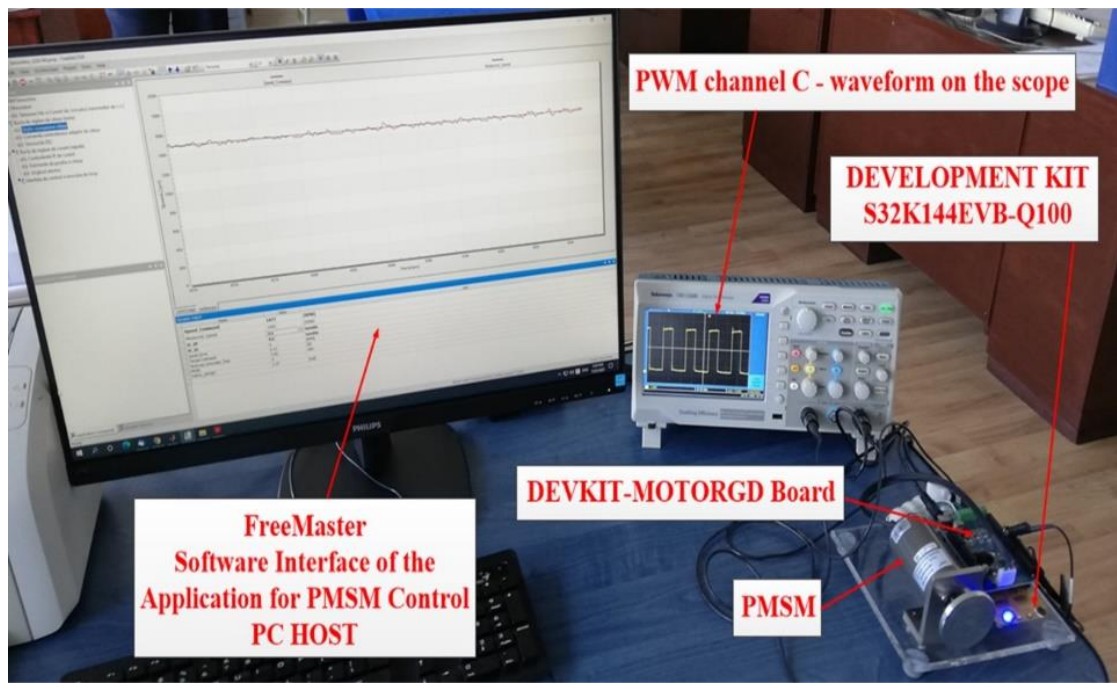

**Figure 33.** Experimental platform image.

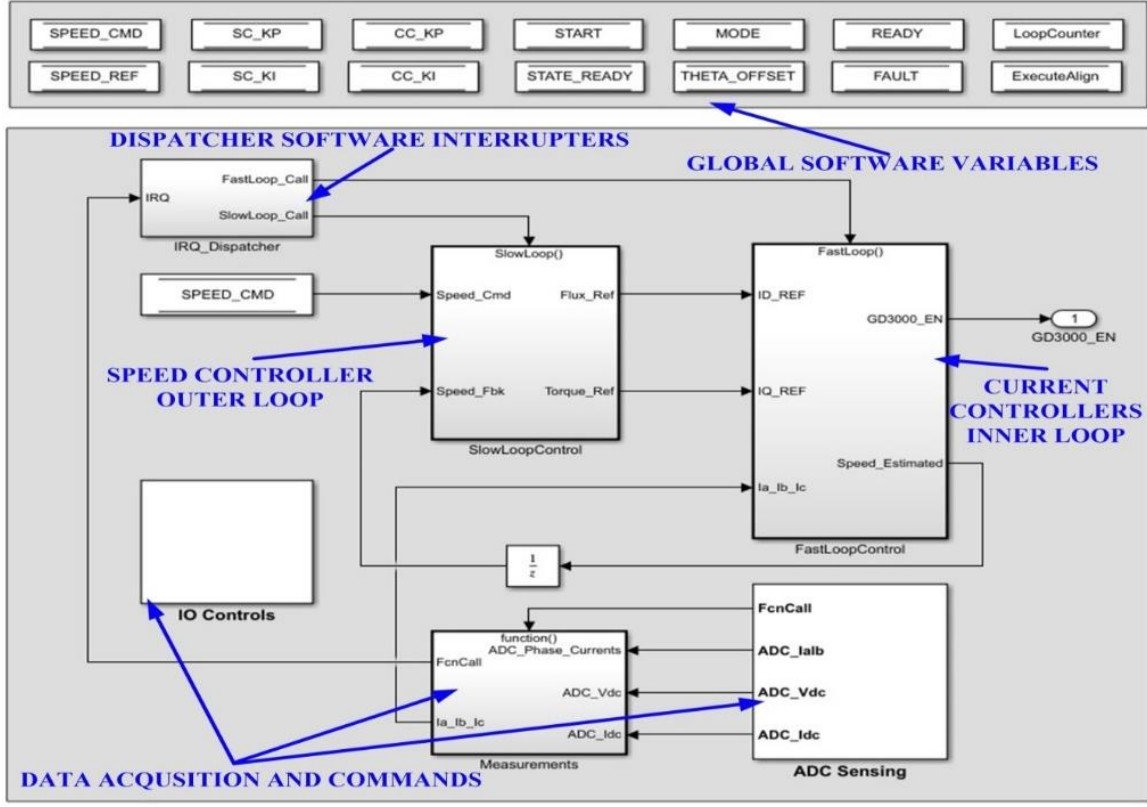

**Figure 34.** Software block diagram of the implementation in MATLAB/Simulink and Model-Based Design Toolbox S32K1xx Series NXP embedded system for PMSM control.

Figure 35 presents the block diagram of the software for the outer speed control loop MATLAB/Simulink model utilizing the bit accurate models for the Automotive Math and Motor Control Library Set for NXP S32K14x devices. The main blocks are: speed reference, initialization speed loop, switching block output command, classical PI speed controller and FO-PI speed controller.

The Discrete Zero-Pole function from Simulink [43] is used for the implementation of the FO-PI speed controller described in Section 4. Equation (108), which represents the discrete form of the FO-PI speed controller, is implemented through this function.

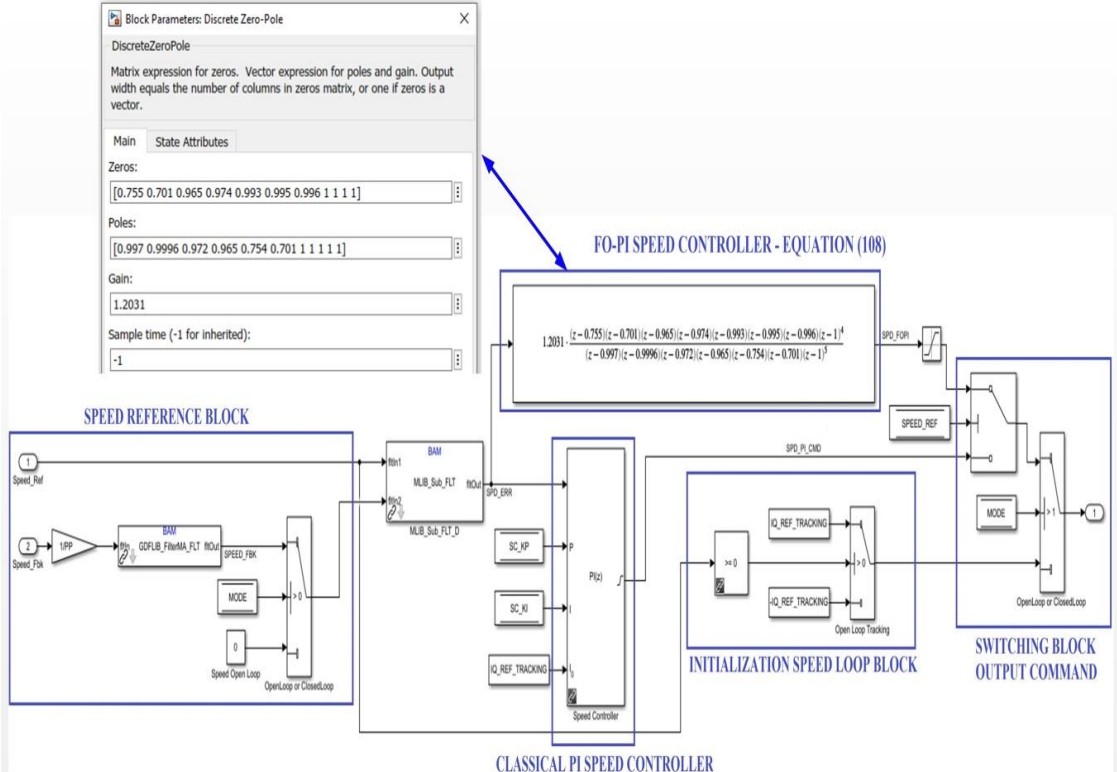

**Figure 35.** Software block diagram of the outer loop speed control MATLAB/Simulink model utilizing the bit accurate models for Automotive Math and Motor Control Library Set for NXP S32K14x devices.

The inner loop for the current control runs every 0.1 ms and the outer loop for the speed control runs every 1 ms. By comparison, each program implemented for numerical simulations, presented in Section 7, runs at each 0.001 ms.

Based on FreeMaster, which is a real-time debugging monitoring software interface for data visualization, configuration and tuning of embedded software applications, the next figure presents the real time evolution of the parameters of the PMSM control system. For reasons of communication between the FreeMaster software interface from the host PC and the controller of the PMSM, the sampling time for the evolution of parameters in Figures 36–44 is 10 ms. Figures 36–38 present the real-time evolution of the PMSM rotor speed with classical PI speed controller.

Figures 39–41 present the real-time evolution of the PMSM rotor speed with FO-PI speed controller. The superior performance of the FO-PI speed controller is clearly noticeable.

The following figures show the real-time evolution of the main PMSM control parameters of interest using the FO-PI speed controller. The real-time evolution of the stator currents are presented in Figure 42, the real-time evolution of $i_d$ and $i_q$ currents are presented in Figures 43 and 44 presents the real-time evolution of the electromagnetic torque.

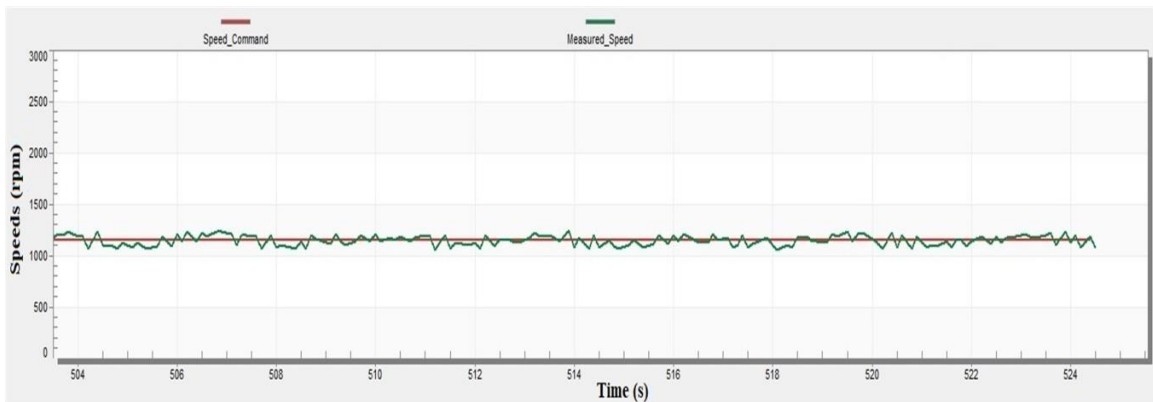

**Figure 36.** Real time evolution of the rotor speed of PMSM with classical PI speed controller for $\omega_{ref}$ = 1150 rpm and no load torque.

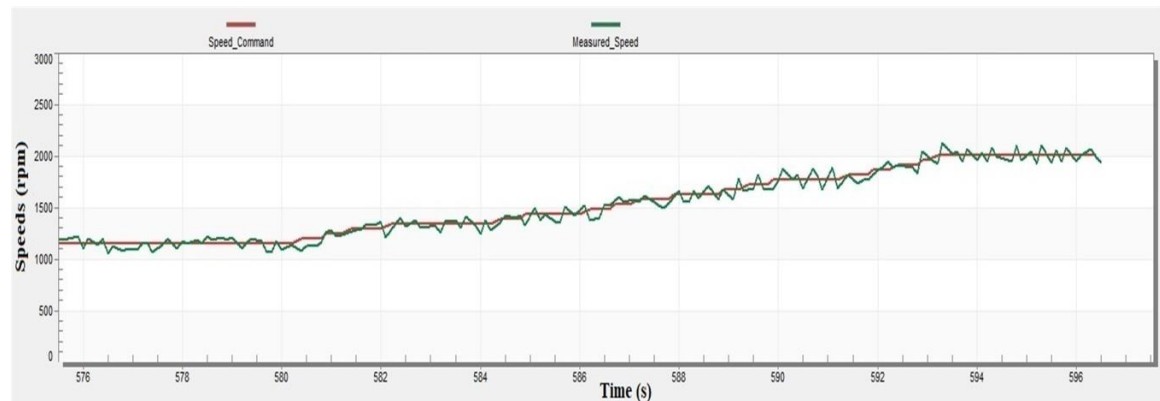

**Figure 37.** Real time evolution of the rotor speed of PMSM with classical PI speed controller for $\omega_{ref}$ between 1150 rpm and 2000 rpm and no load torque.

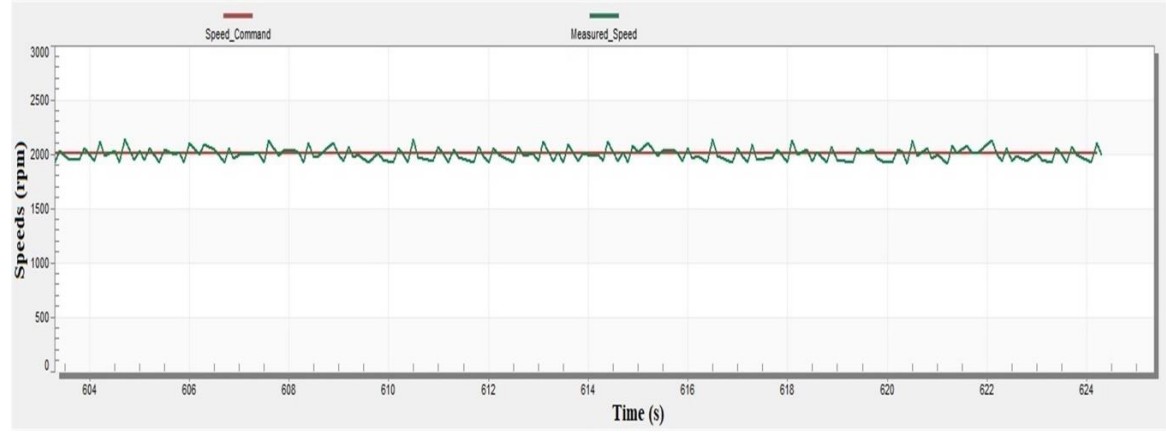

**Figure 38.** Real time evolution of the rotor speed of PMSM with classical PI speed controller for $\omega_{ref}$ = 2000 rpm and no load toque.

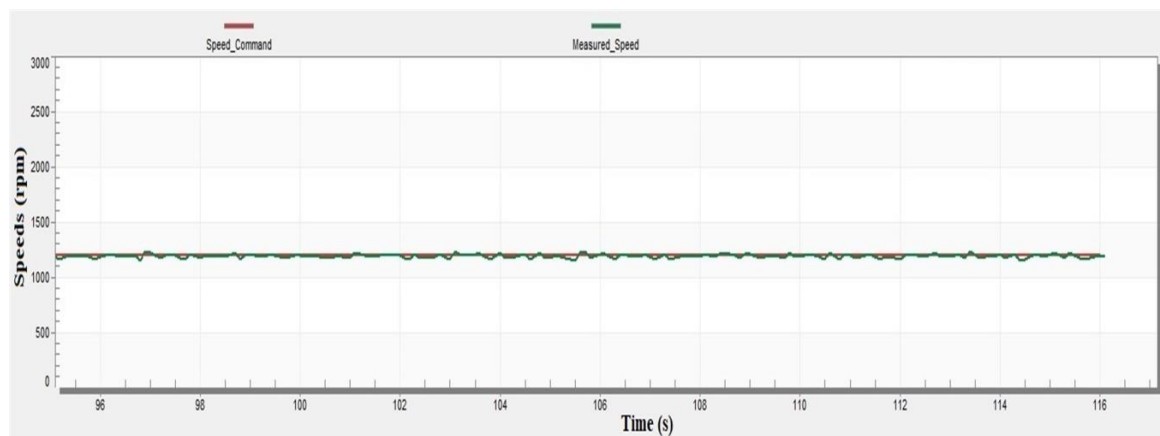

**Figure 39.** Real time evolution of the rotor speed of PMSM with FO-PI speed controller for $\omega_{ref}$ = 1150 rpm and no load toque.

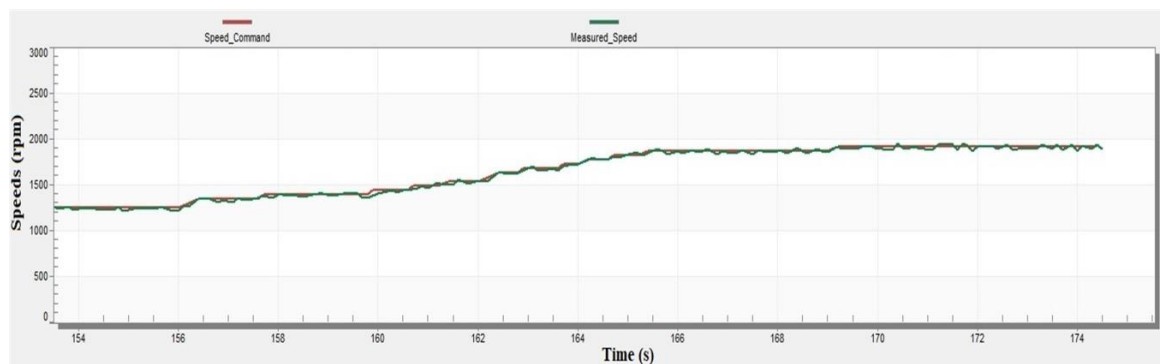

**Figure 40.** Real time evolution of the rotor speed of PMSM with classical FO-PI speed controller for $\omega_{ref}$ between 1150 rpm and 2000 rpm and no load toque.

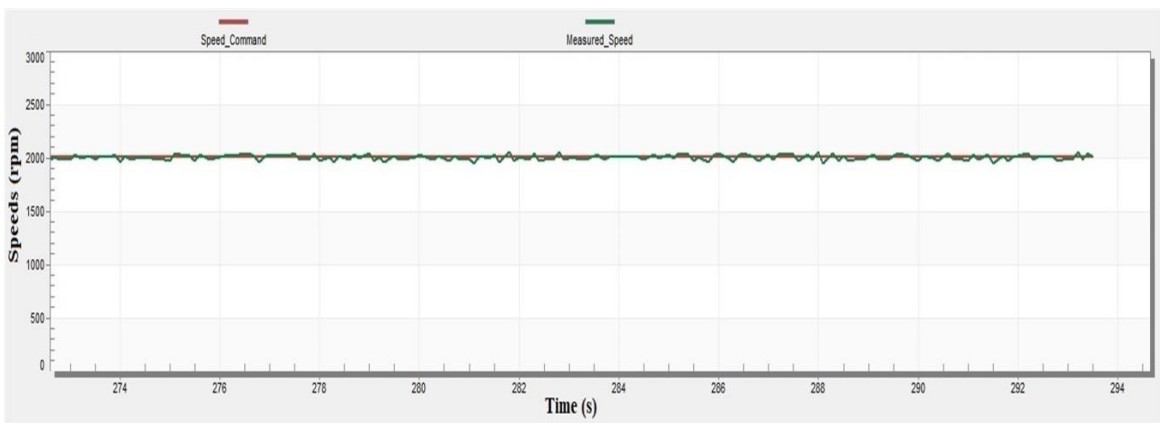

**Figure 41.** Real time evolution of the rotor speed of PMSM with FO-PI speed controller for $\omega_{ref}$ = 2000 rpm and no load toque.

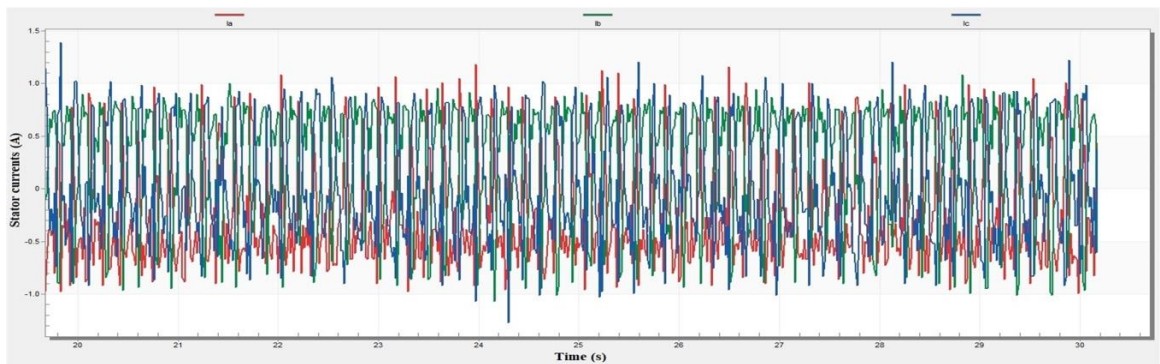

**Figure 42.** Real time evolution of the stator currents of PMSM with FO-PI speed controller.

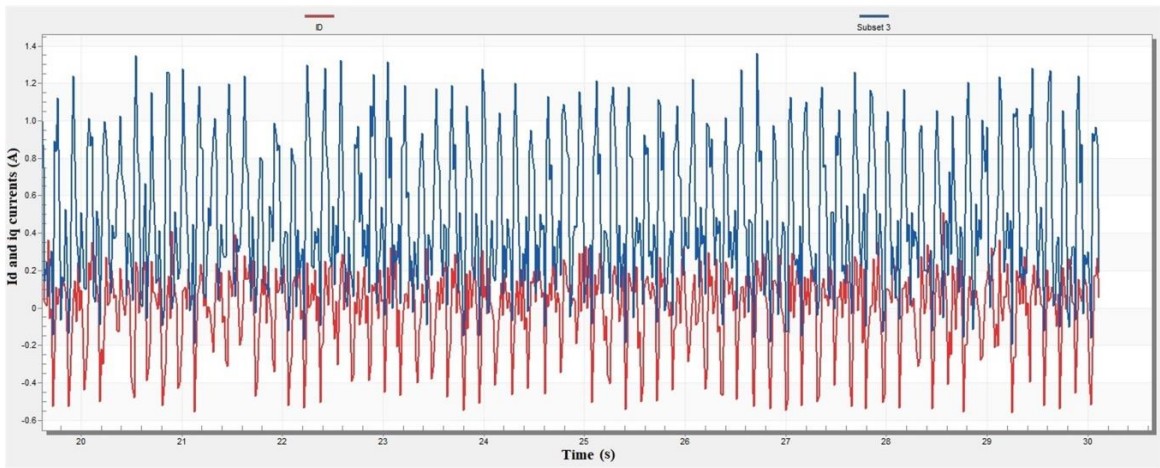

**Figure 43.** Real time evolution of the $i_d$ and $i_q$ currents of PMSM with FO-PI speed controller.

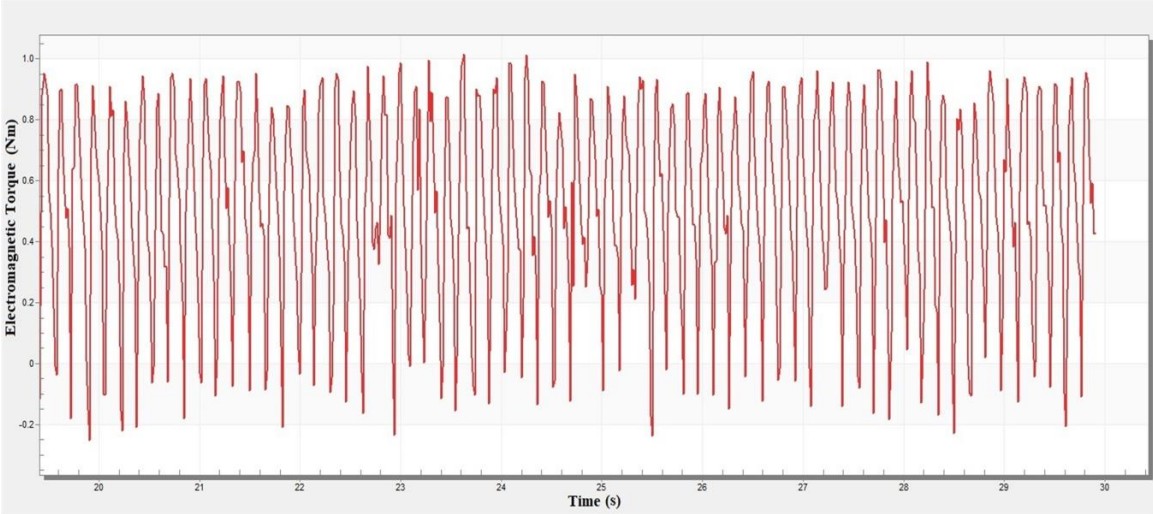

**Figure 44.** Real time evolution of the electromagnetic torque of PMSM with FO-PI speed controller.

It can be noted that there is a similarity between the results obtained by numerical simulation in Figure 13 for the control of the PMSM using an FO-PI speed controller, $\omega_{ref}$ = 2000 rpm and no load torque, and the experimental results presented in Figures 42–44 concerning the stator currents, the electromagnetic torque and $i_d$ and $i_q$ currents. Furthermore, Figures 36–41 show the superiority of the FO-PI speed controller over the classic PI speed controller, in the case of implementation in an embedded system.

## 9. Conclusions

Based on fractional calculus, this article presents a number of fractional controllers for the PMSM rotor speed control loops and $i_d$ and $i_q$ current control loops in the FOC-type control strategy. The proposed system for the PMSM control is based on an FO-SMC speed controller and FO-synergetic current controllers. Due to the additional control parameters generated in the structure of the fractional order controllers, superior performances are obtained for the PMSM rotor speed control. In addition, the sensorless-type PMSM control structure detects the faults of the current sensors and performs a significant reduction in the THD. The parametric robustness of the proposed control system is demonstrated by very good control performances achieved even when the uniformly distributed noise is present in the load torque $T_L$, and under variations by 100% of the load torque $T_L$, of the moment of inertia of $J$ rotor and of the stator resistance $R_s$. The performances of the proposed control system are validated both by numerical simulations and experimentally, through real-time implementation in embedded systems.

**Author Contributions:** Conceptualization, M.N.; data curation, M.N. and C.-I.N.; formal analysis, M.N. and C.-I.N.; funding acquisition, M.N.; investigation, M.N. and C.-I.N.; methodology, M.N. and C.-I.N.; project administration, M.N.; resources, M.N. and C.-I.N.; software, M.N. and C.-I.N.; supervision, M.N. and C.-I.N.; validation, M.N. and C.-I.N.; visualization, M.N. and C.-I.N.; writing—original draft, M.N. and C.-I.N.; writing—review and editing, M.N. and C.-I.N. All authors have read and agreed to the published version of the manuscript.

**Funding:** This paper was developed with funds from the Ministry of Research and Innovation as part of the NUCLEU Program: PN 19 38 01 03.

**Conflicts of Interest:** The authors declare no conflict of interest.

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
