# Peer review of "Sensorless Fractional Order Control of PMSM Based on Synergetic and Sliding Mode Controllers"

_electronics, doi:10.3390/electronics9091494_

Round 1
Reviewer 1 Report
This paper investigates and compares different controllers in the dynamic performance of PMSM using field orient control.
- Please improve the quality of the figures to make them clear for the reader.
- Could you show in a table/figure the robustness of each controller against the parameter variation of the motor/drive system?
- How about the response against the load variation?
- Figure 15 shows that there is no difference between the controllers when the J is high. This may be the real case of the practical system? Then what the benefit of the FO-PI?
- How the controller gains are selected?
- Experimental validation is limited.
Author Response
Dear reviewer, thanks for your recommendations.
- The quality of the figures has been improved (300dpi).
- For this requirement the tables 2, 3 and 4 have been inserted.
- In Figures 13-19 the load torque varies between 0 and 10 Nm.
- Indeed, in the original version, Figures 14 and 15 were wrong. Now they have been redone correctly.
- It was inserted: “The tuning of PI controllers by using Ziegler-Nichols methods is a well-known technique. In the fractional case, the FOMCON toolbox for MATLAB utility program is used for the tuning of FO-PI controllers. In order to obtain the optimal tuning parameters in the fractional case, a number of optimization methods are incorporated in the FOMCON toolbox, both in the frequency range and in the time domain. In the frequency range, the goal of optimizing the parameters is achieved by obtaining optimal performance in terms of the sensitivity function S(jω) for disturbance rejection for low and middle frequency range and the rejection of the high frequency noise using the complementary sensitivity function T(jω). In the time domain, the tuning of fractional controllers is carried out by minimizing optimal criteria, such as the integral absolute error (IAE) [39].”
6. The entire Section 8 has been improved.

Reviewer 2 Report
The authors present in this article a fractional controllers for the PMSM rotor speed control loops, and id and iq current control loops in the FOC-type control strategy. The simulation part is complete, the experimental part must be improved. I have some questions for both simulation and experimental section: 1.- Table 1 shows the nominal parameters of the PMSM. Are the nominal parameters of this table the same used in the experiments in section 8 -->Experimental results ?. 2.- In figure 13 a comparison of the Fo-PI versus PI regulation is made for a reference speed of 300 rpm. And with a sudden variation of the load braking torque it doubles from nominal to double. This variation occurs in a simulation time at t = 0, however around that time it is not detailed, practically it is not seen. On the other hand, a long interval of the permanent regime from 0.3 s to 1 second is represented. Important events are from 0 s to the beginning of the permanent regime which is from 0.3 s. The same can be said of figures 14 and 15, a lot of information is lost from the initial conditions t = 0 until the permanent regime is reached. In my opinion, it should be represented from t = 0 to the beginning of the permanent regime. 3.- In figures 13, 14 and 15. a) What are the parameters used in all the regulators ? . b) What criteria has been followed for the choice of these parameters ? . c) Figures 13, 14 and 15 show the speed variation for the different methods used. But electromagnetic torques and currents do not appear for all the methods used. It is mandatory to analyze how the method affects those quantities that are critical in electrical machines. 4.- In figure 19 a global comparison of the various types of regulators used in the simulations is made, that is: speed FO-SMC controller, speed FO-PI controller, speed TID controller, speed FO-Lead-Lag controller, and speed PI controller - and a ωref = 300rpm, and a braking torque of TL = 10Nm. The parameters used in these simulations are not indicated either in the text or in the figure. a) what has been the criteria for the choice of these parameters of the regulators ?. 5.- Figure 26 explains a comparative time evolution of the numerical simulation for FOC strategy with PI controller, Synergetic controller, and FO-Synergetic controller of the PMSM. a) Which are torque ripple comparison for those methods ? b) And what are the currents ? 6.- In the summary of the article on lines 23 ,24 and 25 it is written:" The performances of the proposed control system are validated both by numerical simulations and
experimentally, through the real-time implementation in embedded systems." In the conclusions lines 636 and 637 it is written:" The performances of the proposed control system are validated both by numerical simulations and experimentally, through the real-time implementation in embedded systems."
The article has an extension of 34 pages and a total of 33 figures related to the numerical simulations. Only one page and one figure are devoted to the experimental part. On that page the Kit used is briefly explained, no reference is made to the implementation of the proposed methods where the results proposed in the article and the classic methods are represented.
The explanation of this part should be expanded.
Figure 34 shows the speed reference versus the measured speed:
a) Which method does it correspond to ?.
b) What are the experimental results of the speeds with the alternative methods seen in the simulations ?.
c) And finally, what are the experimental measurements of the currents, torque, and others of all the methods that correspond to the simulations ?
Thank you very much
Author Response
Dear reviewer, thanks for your recommendations.
- The motor used in Section 8 is Linix 45ZWN24-40 PMSM type and has similar parameters with Table 1.
- Indeed, the chosen simulation time was too long. In this sense, the figures have been remade.
3 and 4. The Figures 13, 14, 15, and 19 have been remade. It was inserted: “The tuning of PI controllers by using Ziegler-Nichols methods is a well-known technique. In the fractional case, the FOMCON toolbox for MATLAB utility program is used for the tuning of FO-PI controllers. In order to obtain the optimal tuning parameters in the fractional case, a number of optimization methods are incorporated in the FOMCON toolbox, both in the frequency range and in the time domain. In the frequency range, the goal of optimizing the parameters is achieved by obtaining optimal performance in terms of the sensitivity function S(jω) for disturbance rejection for low and middle frequency range and the rejection of the high frequency noise using the complementary sensitivity function T(jω). In the time domain, the tuning of fractional controllers is carried out by minimizing optimal criteria, such as the integral absolute error (IAE) [39].”
- Figure 26 has been remade and Table 3 was inserted.
6. The entire Section 8 has been improved.

Reviewer 3 Report
This manuscript presents a fractional-order-based control strategy for PMSMs. However, this manuscript is awfully prepared and lacks clear contributions to the related literature. The specific comments may be of help to improve the quality.
- The English writing is really poor so that it is difficult to grasp the main points the authors want to convey. An English native speaker with relevant technological background is recommended to come with help.
- The figures need to be reformulated to make them more informative.
- The major contributions of this study should be highlighted. To the end, a thorough and complete review should be consciously presented.
- The fractional-order calculus is prohibitively heavy for online implementation. Please give more details on how to ensure real-time implementation.
- A comparative study to the state-of-the-art methods is needed to verify the superiority of the proposed method.
Author Response
Dear reviewer, thanks for your recommendations.
- The whole article was revised from the point of view of English grammar.
- All figures have been improved both qualitatively (300 dpi) and in terms of information provided.
- Indeed it is very difficult to implement a fractional calculation online and to the best of our knowledge there aren’t libraries to automatically translate of the fractional transfer function into directly executable code in a DSP.
In our case: “For the implementation in DSP, it is necessary to obtain the transfer function of the FO-PI controller given in equation (105) in an equivalent form as discrete variable z, but of integer order. For this, according to those presented in Section 2, an approximation of the fractional order transfer function can be obtained with an integer-order continuous transfer function, by using the Oustaloup filter. In the usual frequency range for the presented application ω=[10-2; 103] rad/s, in equation (107) is expressed the equivalent transfer function obtained. In order to obtain the discrete form of this equivalent transfer function, the Tustin substitution is used, and the obtained discrete transfer function is expressed in equation (108).”
“Figure 35 presents the block diagram of the software for the outer speed control loop Matlab/Simulink model utilizing the Bit Accurate Models for Automotive Math and Motor Control Library Set for NXP S32K14x devices. The main blocks are: speed reference, initialization speed loop, switching block output command, classical PI speed controller, and FO-PI speed controller. The Discrete Zero-Pole function from Simulink [40] is used for the implementation of the FO-PI speed controller described in Section 4. Equation (108), which represents the discrete form of the FO-PI speed controller, is implemented through this function.”
3 and 5. Indeed, the required clarifications are necessary. In this sense, a series of such specifications were introduced in the article (the initial version has 9178 words and the current version has 10741 words). Also, the references have been updated (20% are from the "Electronics Journal" 2018-2020) and the vast majority are 2016-2020. The entire Section 8 has been improved.
“This article compares the performances obtained using FO-PI, tilt integral derivatives (TID), FO-Lead Lag controller, and SMC speed controllers against the classic PI-type speed controller in a FOC-type control structure of the PMSM, under the conditions where the controller of the current loops is of PI type. It also presents the performances obtained by using the synergetic control for the control of currents id and iq, within a FOC-type control structure of the PMSM with PI speed controller. The main contribution consists in proposing a PMSM control structure, where the controller of the outer rotor speed control loop is of FO-SMC type, and the controllers for the inner control loops of id and iq currents are of FO-Synergetic type. Superior performances are obtained by using the control system proposed, even in the case of parametric variations. The FO-SMC controller outputs the current reference iqref, while idref=0 according to the FOC control strategy. The FO-Synergetic-type controllers directly provide the control inputs ud and uq, and the control of the inverter is performed through the inverse Park and Clarke transformations from d-q reference frame system to abc reference frame system. The validation of the results presented is achieved by numerical simulations, but also by real-time implementation in embedded systems”.
“It is obviously noticeable that the values obtained for the settling time are of 0.92ms under nominal parameters of the PMSM, along with the other performance indices which can be considered as very good for the FO-SMC speed controller and the FO-Synergistic current controller. The PMSM used in these numerical simulations is implemented in Power Systems/Simscape Electrical toolbox from Simulink, and in many scientific papers it is used as benchmark, and, to our best knowledge, the settling time of 0.92ms obtained when using the FO-SMC speed controller and the FO-Synergistic current controller is the best settling time obtained for a usual range of the speed reference and load torque, and for any other proposed controllers”.

Round 2
Reviewer 1 Report
Thank you.
Kind regards,
Author Response
Dear reviewer, thanks for your recommendations.
In the introduction we inserted the following fragment: “Among the usual applications of the PMSM control systems we mention: maintaining the speed according to a profile set by a speed reference generator, but also master/slave type multi-motor applications where the coupling is rigid or flexible and it is necessary to maintain the same speed or maintain the torque developed by each engine in the narrowest range possible [36]. Also, the electric vehicles drive control applications raise the problem of multi-motor speed control [37]. Applications such as the automatic control of the hydropower dam spillway require that the error accumulated in each drive chain corresponding to each engine be less than the set value [38]. These applications are generally achieved using the controllers described above, but of the integer order type. In this article we will focus on the fractional order controllers which provide superior control performance, but also on the increased difficulties regarding the implementation in embedded systems.”
The whole article was revised from the point of view of English grammar.
All figures have been improved both qualitatively (300 dpi) and in terms of information provided.
Reviewer 2 Report
The authors have notably improved the experimental section of the article,and have responded to the questions asked. In the same way, they have clarified the rest
of the questions raised in the simulations section.
In my opinion the article can be published in the current version. Thank you very much
Author Response
Dear reviewer, thanks for your assessments.
Reviewer 3 Report
This manuscript presents fractional-order controllers for PMSMs. However, there are still many places that need further revisions.
- The introduction needs further revisions. The industrial applications of PMSMs should be provided to improve the completeness and comprehensiveness. For example, Battery aging assessment for real-world electric buses based on incremental capacity analysis and radial basis function neural network; Thermal runaway behavior during overcharge for large-format Lithium-ion batteries with different packaging patterns.
- The writing is awful. There are plenty of grammar errors and typos throughout the manuscript. Please carefully read proof the manuscript. It is recommended that an English native speaker with relevant technological background comes to help with writing.
- There are too many figures in the manuscript. Please just provide the most relevant figures to inform the main findings of this study. Besides, the resolutions need improvement to make them more clear and informative. The font sizes and styles should be the same for all the figures.
- The conclusions should be enriched by including the exclusive contributions of this study. The accurate experimental results and the improvement relative to other methods should be provided in details to exhibit the superiority of the proposed method.

Author Response
Dear reviewer, thanks for your recommendations.
Attached to this reply is a list of scientific papers published by the authors (Nicola et al.), in which almost all the types of controllers and observers presented in the introduction are implemented. These refer to integer order controllers. In this article we present the comparisons between the types of fractional order controllers, together with the calculi and the demonstration of their stability, simulations and implementation in DSP. Also, the list attached to articles [14, [15], and [16] presents applications in the mining industry, electrical vehicles control, and the automatic control of the hydropower dam spillway.
We specify that we have also made a series of reviews for articles in IEEE Xplore conferences, but also for Scopus journals and IEEE Access journal, therefore we understand quite well the expectations of the editor and of the readers. In this case, the article is to be published in “Electronics/Electrical and Autonomous Vehicles/Advanced Control Systems for Electric Drive”, hence at this level the reader understands very clearly the possible applications of the advanced control systems presented, and additional details are considered redundant.
In the introduction we inserted the following fragment: “Among the usual applications of the PMSM control systems we mention: maintaining the speed according to a profile set by a speed reference generator, but also master/slave type multi-motor applications where the coupling is rigid or flexible and it is necessary to maintain the same speed or maintain the torque developed by each engine in the narrowest range possible [36]. Also, the electric vehicles drive control applications raise the problem of multi-motor speed control [37]. Applications such as the automatic control of the hydropower dam spillway require that the error accumulated in each drive chain corresponding to each engine be less than the set value [38]. These applications are generally achieved using the controllers described above, but of the integer order type. In this article we will focus on the fractional order controllers which provide superior control performance, but also on the increased difficulties regarding the implementation in embedded systems.”
The whole article was revised from the point of view of English grammar.
All figures have been improved both qualitatively (300 dpi) and in terms of information provided. As opposed to the initial version, new figures were inserted for the requirements of reviewer 1 and reviewer 2.
List of papers about control of PMSM (Nicola et al.) – IEEE Xplore
- Sensorless Control of PMSM using FOC Strategy based on LADRC Speed Controller, Proceedings of the 12th Edition Electronics, Computers and Artificial Intelligence (ECAI 2020), Bucuresti, Romania, 25-27 June, 2020, pp. 1-6;
- Sensorless Control of PMSM using Backstepping Control and ESO-type Observer, Proceedings of the 12th Edition Electronics, Computers and Artificial Intelligence (ECAI 2020), Bucuresti, Romania, 25-27 June, 2020, pp. 1-6;
- Sensorless Control of PMSM using FOC Strategy Based on PI-ILC Law and Sliding Mode Observer, Proceedings of the XXIst International Symposium on Electrical Apparatus and Technologies (SIELA 2020), Bourgas, Bulgaria, 3-6 June, 2020, pp. 1-6, DOI: 1109/SIELA49118.2020.9167046;
- Sensorless Control of PMSM using DTC Strategy Based on Multiple ANN and Load Torque Observer, Proceedings of the XXIst International Symposium on Electrical Apparatus and Technologies (SIELA 2020), Bourgas, Bulgaria, 3-6 June, 2020, pp. 1-6, DOI: 1109/SIELA49118.2020.9167120;
- Sensorless Control of PMSM using FOC Strategy Based on Multiple ANN and Load Torque Observer, Proceedings of the 15th International Conference on Development and Application Systems (DAS), Suceava, Romania, 21-23 May, 2020, pp. 32-37, DOI: 10.1109/DAS49615.2020.9108914;
- Sensorless Control of PMSM using DTC Strategy Based on PI-ILC Law and MRAS Observer, Proceedings of the 15th International Conference on Development and Application Systems (DAS), Suceava, Romania, 21-23 May, 2020, pp. 38-43, DOI: 10.1109/DAS49615.2020.9108974;
- Identification and Sensorless Control of PMSM Using FOC Strategy and Implementation in Embedded System, Proceedings of the 12th International Conference and Exhibition on Electromechanical and Energy Systems (SIELMEN), Chișinău, Moldova, 10-11 October, 2019, pp. 335-340; DOI: 10.1109/SIELMEN.2019.8905893;
- Power Factor Correction and Sensorless Control of PMSM Using FOC Strategy, Proceedings of the 12th International Conference and Exhibition on Electromechanical and Energy Systems (SIELMEN), Chișinău, Moldova, 10-11 October, 2019, pp. 329-334, DOI: 10.1109/SIELMEN.2019.8905841;
- Sensorless Control for PMSM Using Model Reference Adaptive Control and back-EMF Sliding Mode Observer, Proceedings of the 12th International Conference and Exhibition on Electromechanical and Energy Systems (SIELMEN), Chișinău, Moldova, 10-11 October, 2019, pp. 317-322, DOI: 10.1109/SIELMEN.2019.8905805;
- Sensorless Predictive Control for PMSM Using MRAS Observer, Proceedings of the 12th International Conference and Exhibition on Electromechanical and Energy Systems (SIELMEN), Chișinău, Moldova, 10-11 October, 2019, pp. 323-328, DOI: 10.1109/SIELMEN.2019.8905815;
- Adaptive Sensorless Control of PMSM using Back-EMF Sliding Mode Observer and Fuzzy Logic, Electric Vehicles International Conference (EV2019), București, Romania, 3-4 October, 2019, pp 1-6, DOI: 1109/EV.2019.8893070;
- Sensorless Control of Multi-Motors BLDC using Back-EMF Observer, Proceedings of the 8th International Conference on Modern Power Systems (MPS), Cluj Napoca, Romania, 21-23 May, 2019; pp 1-6, DOI: 10.1109/MPS.2019.8759761;
- Delay Compensation in the PMSM Control by using a Smith Predictor, Proceedings of the 8th International Conference on Modern Power Systems (MPS), Cluj Napoca, Romania, 21-23 May, 2019, pp 1-6, DOI: 10.1109/MPS.2019.8759752;
- Simulation and Implementation of Sensorless Control in Multi-Motors Electric Drives with High Dynamics, Advances in Science, Technology and Engineering Systems Journal (ASTEJ), Vol. 2, No. 4, 2017, pp. 59-67, USA, ISSN: 2415-6698, DOI: 10.25046/aj020409;
“In this article we'll tackle the control of multi-motors electric drives with high dynamic, with rapid changes in torque and speed, with rigid or flexible coupling of motors, where the control strategy is FOC (Field Oriented Control) for each drives and the distributed control in local network using the CANopen protocol. In the surface mining industry, from which the electric drive application for this article is selected, the general trend is toward using asynchronous motors with short-circuit rotor, due to the advantages of this motor both in terms of design and operation. In order to achieve the variable speed, must be used the static frequency converters with sensorless control, where speed is estimated using a Model References Adaptive Control Estimator. The global control system proposed in this paper contain this type of MRAC estimator together with PI-control based, who ensures a good dynamic performance but in a lower complexity of structure such that are properly to implement in real time in a distributed control system with DSP in local network using the CANopen protocol with advantages in terms of software technology, as well as control cost and flexibility of use. Following these directions a functional application was implemented and tested in practice.”
- Sensorless Control of Multi-Motors PMSM using Back-EMF Sliding Mode Observer, Proceedings of the Electric Vehicles International Conference (EV2019), București, Romania, 3-4 October, 2019, pp 1-6, DOI: 1109/EV.2019.8892950;
“This article presents a sensorless control system based on back-EMF (back electromotive force) sliding mode observer for control of rigid coupling of PMSM multi-motors. The back-EMF eα and eβ are provided by a sliding mode observer in order to estimate the rotor speed and position. The main advantage of the sensorless control consists in eliminating the speed transducers thus contributing to the increased reliability of the entire system. The control structure presented is cascaded, following the FOC (Field Oriented Control) principles, where the inner current loop has the reference Id set to be equal to zero, and the Iq reference is set by the PI-type speed controller in the outer loop. Considering that the load torque distributed to the two motors can be balanced or unbalanced, the main task of the multi-motor PMSM sensorless control system is to maintain the same speed of the slave system and the master system and that both systems match the required speed profile. Real time implementation of the multi-motor PMSM sensorless control system is recommended by the good results obtained from numerical simulations.”
- Automatic Control of a Hidropower Dam Spillway, Proceedings of the International Conference on Applied and Theoretical Electricity (ICATE), Craiova, Romania, 8-9 October, 2010, published in Annals of the University of Craiova - Electrical Engineering Series, No. 34, Issue 1, 2010, ISSN 1842-4805.
“A very important element for the safe operation of a hydropower dam consists in the possibility to quickly and safely control the lappet and the gate of dam. This paper presents an application concerning the automatic control and monitoring of a mechanical, electrical and hydraulic plants performing the operation of the penstock mounted on a hydropower dam. The steps to be taken are: the design, control, automation of the electric power plant and the design of the protection, the integration of the overall operation of the whole dam spillway in SCADA system, the purchase, the manufacturing of the cabinets and panels, the mounting of the equipment, the completion of power circuits, secondary circuits, communication and finally testing and commissioning. The role of the automatic control consists in the increase of the safety of the entire dam. The difference imposed between the left and right driving chain of a gate is less than 1 cm. In addition, a complex control system is achieved on four levels, which performs a balanced opening and closing of the gates for a controlled discharge. This paper presents the general architecture of the control system, the algorithm, the communication and the general performance. The authors of this paper have achieved the hardware design and software implementation.”
“Several data of the application (for two dams) could provide a better image about its complexity 2 Workstations, 13 PLC, 13 Operator panel, 24 Communication networks, 23 Automation loops, 16 Static frequency converters, 23 Network analyzers 100 programmable interfaces and transducers, about 5000 variables in 28 software programs.”